# Navigating the Power of Artificial Intelligence in Risk Management: A Comparative Analysis

Mohammad Yazdi [1,2,*] , Esmaeil Zarei [3,4] , Sidum Adumene [5] and Amin Beheshti [6]

1   School of Computing, Engineering & Physical Sciences, University of the West of Scotland (UWS),
    London E14 2BE, UK
2   School of Engineering, Faculty of Science and Engineering, Macquarie University,
    Sydney, NSW 2109, Australia
3   Department of Safety Science, College of Aviation, Embry-Riddle Aeronautical University,
    Prescott, AZ 86301, USA; zareie@erau.edu
4   Robertson Safety Institute (RSI), Embry-Riddle Aeronautical University, Prescott, AZ 86301, USA
5   School of Ocean Technology, Fisheries and Marine Institute, Memorial University of Newfoundland,
    St. John's, NL A1C 5R3, Canada; sadumene@mun.ca
6   Centre for Applied Artificial Intelligence, Macquarie University, Sydney, NSW 2109, Australia;
    amin.beheshti@mq.edu.au
*   Correspondence: mohammad.yazdi@mq.edu.au

**Abstract:** This study presents a responsive analysis of the role of artificial intelligence (AI) in risk management, contrasting traditional approaches with those augmented by AI and highlighting the challenges and opportunities that emerge. AI, intense learning methodologies such as convolutional neural networks (CNNs), have been identified as pivotal in extracting meaningful insights from image data, a form of analysis that holds significant potential in identifying and managing risks across various industries. The research methodology involves a strategic selection and processing of images for analysis and introduces three case studies that serve as benchmarks for evaluation. These case studies showcase the application of AI, in place of image processing capabilities, to identify hazards, evaluate risks, and suggest control measures. The comparative evaluation focuses on the accuracy, relevance, and practicality of the AI-generated findings alongside the system's response time and comprehensive understanding of the context. Results reveal that AI can significantly enhance risk assessment processes, offering rapid and detailed insights. However, the study also recognises the intrinsic limitations of AI in contextual interpretation, advocating for a synergy between technological and domain-specific expertise. The conclusion underscores the transformative potential of AI in risk management, supporting continued research to further integrate AI effectively into risk assessment frameworks.

**Keywords:** artificial intelligence; system safety management; image data analysis; hazard identification; storytelling

## 1. Introduction

Risk management has always been a cornerstone of organisational strategy, forming the bulwark against the system's potential financial, strategic, operational, and reputation losses [1,2]. Its importance lies in its ability to identify, assess, and prioritise the risks, followed by resource allocation to minimise, control, and monitor the probability and/or impact of unfortunate events into an acceptable level of the risks or ALARP "*as low as reasonably practicable*" in some contexts [3–5]. In essence, risk management is not just about averting crises; it is about navigating through them with minimal damage and emerging resilient on the other side.

As we are going into the 21st century, the advent of artificial intelligence (AI) in risk management can mark a significant paradigm shift. AI, with its ability to process

and analyse large volumes of data at speeds and accuracies unattainable by humans, can become a game-changer. It can enhance capability for the identification of potential risks and opportunities with greater precision, thereby enhancing decision-making processes. Machine learning models, especially those harnessing the power of deep learning, are now fundamental tools in predicting market trends, detecting fraudulent activities, and automating risk assessment tasks [6–10].

The progression of AI has led to immersive developments. In today's information-driven world, the multifaceted potential of image data as an invaluable decision-making asset has increasingly caught the attention of industries and researchers alike. Image data provides a granular and rich form of information, capturing subtle patterns often elusive to traditional data forms. Its significance in the domain of risk management cannot be understated. As highlighted by Goodfellow et al. [11], deep learning techniques, especially convolutional neural networks (CNNs), have shown substantial promise in gleaning insights from image data for diverse applications.

Moreover, the challenges in harnessing image data effectively are manifold. High dimensionality, potential biases in data, and the need for substantial computational resources often make image data analysis a daunting task. Moreover, as underscored by Bengio [12], the interpretation and contextual understanding of image data are as critical as the computational aspects, particularly in domains like risk management where stakes are high. Recent advancements in machine learning, particularly deep learning, offer promising avenues for comprehensive image data analysis. For instance, LeCun et al. [13] extensively discussed the role of CNNs in image recognition, which could be repurposed for risk analysis by identifying anomalies or patterns indicative of potential risks. The integration of image data into risk management presents a transformative opportunity. For sectors like high-tech industries, where risks often manifest in numerical indicators, the use of image data (e.g., satellite imagery) can forecast potential economic downturns by analysing changes in physical infrastructure. In health, medical imaging can pre-empt potential outbreaks or diagnose illnesses at a scale and precision previously unattainable. Girshick et al. [14] work on object detection can be a boon for security and surveillance, flagging potential physical threats in real time. Furthermore, Zhou et al. [15] emphasised the need to pair technological advancements with domain-specific knowledge. This integration of tech and expertise amplifies the potential applications of image data in risk management, ensuring insights are both precise and actionable. The dynamic interplay of challenges and advancements paints a promising picture for image data in risk management. By embracing this frontier, industries stand to gain unprecedented depth in risk assessment practices, opening avenues for intelligent proactive and informed strategies.

Therefore, the integration of these advanced AI technologies, among which is Chatbot, like ChatGPT-4 (https://openai.com/ accessed on 2 March 2024), an advanced iteration of generative pre-trained transformers developed by OpenAI, into the domain of risk management not only presents transformative opportunities but also introduces a new era of efficiency and effectiveness. For industries ranging from manufacturing, oil and gas, marine, and finance to healthcare, the capabilities of such AI systems can lead to the development of predictive models that identify potential risks with unprecedented accuracy. These AI-driven insights, when combined with domain expertise, can fortify strategies, enabling organisations to respond to threats with agility and informed confidence [16,17]. In recent studies, the application and accuracy of ChatGPT in risk management for construction projects have been extensively examined. Aladağ (2023) [18,19] highlights ChatGPT's capability to deliver accurate risk-based decisions across various project types, emphasising the importance of key performance indicators (KPIs) in risk management. Hofert (2023) [20] provides an in-depth analysis of ChatGPT's proficiency in quantitative risk management, offering a numerical assessment of its performance and effectiveness in managing project risks. Furthermore, Al-Mhdawi et al. (2023) [19] focus on evaluating ChatGPT's performance in the risk management process as per the ISO 31000 standard, showcasing its potential in managing construction risks effectively.

Lastly, Nyqvist, Peltokorpi, and Seppänen (2024) [21] explore the capabilities of the Chat-GPT GPT-4 model against human experts in construction project risk management, aiming to ascertain if AI can surpass human proficiency in this domain. These studies collectively underscore the evolving role of AI tools like ChatGPT in enhancing the precision and effectiveness of risk management strategies in the construction industry. Building upon the foundational research on ChatGPT's application in construction risk management, recent scholarly works have broadened the exploration of its capabilities and regulatory needs in various fields. Hacker, Engel, and Mauer (2023) [22] discuss the necessity for a comprehensive risk management framework for large generative AI models like ChatGPT, emphasising the need to address potential high-risk applications and innate vulnerabilities of these technologies. In the realm of finance and accounting, Rane (2023) [23] investigates the role and challenges of ChatGPT and similar AI systems, pointing out the critical need for secure and dependable financial management, while also highlighting AI's potential in risk management within these sectors [18,24,25]. He et al. [26] explore the critical role of AI in transforming project risk management, a key determinant of success in modern organisations. They examine how AI revolutionises risk identification, evaluation, and mitigation, thereby improving project outcomes. The chapter details how predictive analytics and machine learning mine historical data for patterns, enabling proactive risk management and simulation-driven approaches.

Accordingly, we would like to draw the attention of our esteemed readers to a poignant instance from the past: the 2016 Brussels airport bombings. This tragic event underscores the significant impact that enhanced risk management in image data analysis could have in averting disasters. Specifically, it demonstrates the potential of using AI to identify suspicious activities and unattended items. By analysing image data and integrating it with additional intelligence, AI systems can create comprehensive narratives that provide vital context and insights for security personnel. Such advanced warning systems empower rapid and effective responses to emerging threats, which could prevent harm and save lives. Figure 1 illustrates an end-to-end risk management process using image data analysis. It emphasises the potential of harnessing camera footage from airports to enhance our comprehension of events and activities.

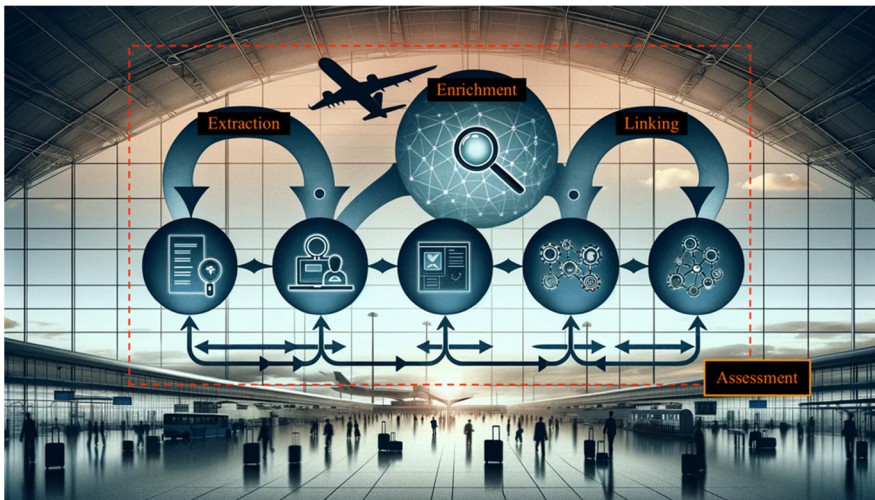

**Figure 1.** The end-to-end risk management process using image data analysis (modified after AI-generated).

By analysing details from this image, an AI system can specify potential hazards and construct comprehensive risk management that offers valuable context for security personnel. Integrating these data with external knowledge sources, such as social databases and historical police records, the AI system further enhances the risk management depth. This comprehensive understanding allows the system to spotlight potential red flags, identify

imminent threats, and offer actionable recommendations for response. As a language model with image input capabilities, such AI tools are not just a step up from their predecessors in understanding and generating text; they are also capable of interpreting visual data, thereby broadening their applicability across various domains. Its understanding of context, ability to process complex instructions, and generate human-like text while also being able to analyse image data make it particularly useful in risk management. We believe that by utilising its advanced capabilities, AI tools can play a pivotal role in hazard identification in diverse workplaces even by sifting through social media and other digital platforms to pinpoint potential safety concerns. It can also analyse satellite imagery to identify environmental hazards, thereby aiding in the early detection and prevention of ecological threats. In the area of workplace health and safety (WHS), they can be instrumental in scrutinising workplace imagery to uncover safety issues (e.g., simple photo shut or panorama), contributing to the enhancement of WHS protocols. Moreover, the interpretation of medical imagery can assist in the early detection of health conditions, facilitating timely interventions [27]. Beyond mere identification, they can also support the development and implementation of targeted control measures, ensuring comprehensive risk mitigation strategies are in place. This not only enhances proactive safety management but also ensures that risks are managed with an informed and responsive approach.

The field of risk management is evolving, driven by technological advancements in AI and the ever-growing capabilities of models like Chatbots. As we navigate through the intricacies of managing risks in an information-driven age, the synergy between AI's analytical might and human expertise becomes ever more crucial. Accordingly, in this study, our discourse aims to dissect and understand this synergy and explore both the transformative potential and the attendant challenges of leveraging image data for robust risk management strategies [28–31]. Based on the given context and the objectives of the study, three main research questions that could guide the discourse are as follows:

- How do AI technologies enhance the identification and assessment of risks in various industries? This question explores the capabilities relevant to risk management, such as its ability to process and analyse large datasets, including image data, and how these capabilities can be leveraged to improve hazard identification, risk assessment, and decision-making processes within different sectors.
- What synergies and potential challenges emerge when integrating human expertise with AI-driven risk management strategies? This question seeks to understand the interplay between human intuition, experience, domain knowledge, and the data-driven insights provided by AI. It also considers the potential barriers to integration, such as resistance to change, interpretability of AI decisions, data privacy concerns, and the need for appropriate governance frameworks.
- In what ways does the utilisation of image data, analysed by AI systems, lead to the development of proactive and preventive risk management measures across industries? The focus here is on the predictive power of AI when applied to image data and storytelling, examining the types of control measures that AI analyses can inform. Additionally, it explores the implications for early detection and prevention of risk occurrences, ultimately guiding industries toward more proactive risk management practices.

The structure of the rest of the paper is outlined as follows: Section 2 discusses the role of artificial intelligence in risk management. Section 3 investigates three different case studies through a comparative analysis. In Section 4, a comparative evaluation is conducted. Finally, the paper concludes with a summary of key findings and concluding remarks in the last section.

## 2. The Role of AI in Risk Management

As mentioned in the introduction, considering the industrial and academic experience and supported by the state of the arts, the field of risk management has traditionally revolved around the identification, analysis, and prioritisation of potential risks, followed

by the coordinated application of resources to minimise, monitor, and control the risk of the unfortunate events. Traditional methods rely heavily on historical data and human expertise, with tools like risk registers, risk matrices, failure model and effect analysis (FMEA) [32–34], checklists, and qualitative assessments being predominant [35–37]. However, the advent of AI-enhanced risk management is a paradigm shift. AI systems, with their superior computational capabilities, can analyse vast quantities of data, uncover patterns invisible to the human eye, and predict future risks with higher accuracy. By juxtaposing traditional risk management with AI-enhanced risk management, we can observe a significant transformation characterised by real-time data processing, predictive analytics, and the ability to adapt to new information.

AI introduces specific capabilities that dramatically enhance risk identification and evaluation processes. Machine learning and data mining provide sophisticated means to analyse large datasets for predictive insights, detect anomalies, and forecast trends. Machine learning algorithms can be trained to recognise complex patterns and make decisions with minimal human intervention [38]. For risk evaluation, AI systems utilise these patterns to predict potential risks and their impacts, enabling organisations to allocate their resources more efficiently. Using these technologies in risk management not only can accelerate the decision-making process but also improve its accuracy, which is paramount in critical risk-laden scenarios.

Chatbot, i.e., GPT4 (from here called AIc), an advanced iteration of the Generative Pre-trained Transformer models, has sophisticated image processing capabilities. It incorporates a deep learning architecture capable of interpreting visual data, drawing inferences, and generating contextual information. This ability is particularly beneficial for risk management applications where visual data, such as images from surveillance cameras or satellite imagery, play a vital role. AIc processes this data through CNNs, adept at handling pixel-based information and extracting features essential for image classification, object detection, and semantic segmentation.

AIc's interpretation of images is a multi-faceted process involving several stages. Initially, the model pre-processes the image data to normalise and structure it for analysis. Then, using its neural network, AIc identifies various elements within the images, recognising objects and discerning patterns [39]. It can analyse the context within the image, connecting it to its vast knowledge base for comprehensive understanding. Despite its advanced capabilities, AI image recognition has its nuances and limitations. One of the primary concerns is the quality and bias of the training data [40,41]. Suppose the data used to train the model must be more diverse and comprehensive. In that case, the AI system may develop biases or be unable to accurately recognise specific scenarios, leading to potentially significant consequences in risk management situations. Additionally, AI systems may need help interpreting images that contain novel elements or are of low quality, which can result in inaccuracies. The existing literature, as referenced by [25], critiques that AIcs often appear to be "*jack of all trades, master of none*". This suggests that while AIcs can perform a wide range of tasks, they may lack depth in any single area of expertise. Consequently, it is crucial for risk assessors to thoroughly evaluate the comprehensiveness and accuracy of AIcs. This rigorous assessment is necessary to ensure that AIcs are not only versatile but also reliable and proficient in their responses and functionalities.

In this study, a stringent set of criteria guided the selection and processing of images for this study to ensure relevance and accuracy. Images were chosen based on their applicability to the risk management scenarios under investigation and were pre-processed to enhance quality and uniformity. This pre-processing included steps such as resolution normalisation, contrast adjustments, and removing irrelevant metadata, thereby facilitating more accurate analysis by the AI system. The methodology incorporates three case studies, each selected for its unique perspective on the application of AI in risk management. These case studies span different industries and risk types, from operational to strategic risks, providing a holistic view of AI's potential in this domain. Each case study will detail

the risks involved, the AI's role in managing those risks, and the outcomes relative to traditional risk management approaches.

*Methodology*

The methodology section of the study provides a comprehensive examination of how AI, notably through machine learning and data mining, can augment traditional risk management practices. Traditional methods, which depend on historical data and human expertise, utilise tools such as risk registers and matrices along with qualitative assessments. The AI-driven approach, however, marks a significant shift by utilising sophisticated algorithms to parse through extensive datasets, discern intricate patterns, and forecast potential risks with greater precision.

The focus is primarily on the application of AI technologies, including machine learning and data mining, to bolster risk identification and evaluation efforts. These cutting-edge technologies are adept at scouring through sizable datasets to extract predictive insights, detect anomalies, and project trends, thus considerably enhancing the speed and accuracy of decision-making in risk management.

Illustrating AI's application in risk management, the study discusses the use of advanced AI models like GPT-4 (AIc), renowned for their refined image processing abilities. AIc employs convolutional neural networks (CNNs) to interpret visual data from sources such as surveillance footage and satellite images, an indispensable capability in risk scenarios where quick and precise image analysis is paramount.

The methodology unfolds as a multi-stage process. It begins with the preprocessing of image data by AIc, which includes standardisation and normalisation techniques to prepare images for analysis. Following this, AIc deploys algorithms for pattern recognition and contextual interpretation within the images. Throughout this process, the study recognises and addresses potential challenges such as data quality and bias, which necessitates a careful evaluation of AIc's comprehensiveness and precision.

The research methodology adopts a systematic approach to selecting and preprocessing images pertinent to risk management scenarios. The study presents three case studies across various industries to illustrate the versatile potential of AI in risk management contexts. Each case study contrasts AI's performance in identifying and managing risks against traditional methodologies, providing empirical evidence of AI's efficacy.

A detailed example within the methodology is the "Annual Airline Gala" case study, where AIc's algorithms are tasked with identifying potential workplace health and safety (WHS) hazards. In this imaginative scenario, the study meticulously documents AIc's operational sequence, including the explicit commands used for image processing, the algorithm parameters set for hazard detection, and the computational processes employed to interpret the visual data. This scenario is not only designed to evaluate AIc's risk assessment capabilities but also its alignment with regulatory standards.

The methodology section outlines every step of the AI operations. For the preprocessing stage, the commands and parameters for image manipulation are enumerated, such as cropping, resizing, and filtering techniques that facilitate enhanced AI analysis. The model training phase involves detailing the architecture of the neural networks used, the training datasets, the validation steps to prevent overfitting, and the testing procedures to measure the model's performance. Each of these steps is described with sufficient granularity to allow for precise replication in future research endeavours.

Furthermore, the details of the software and tools, including version numbers and configurations, were used throughout the study. This ensures that other researchers can replicate the conditions under which the AIc system was evaluated. By doing so, the work provides clarity on the AI's operational framework and also sets a standard for methodological transparency and reproducibility in AI-driven risk management research.

### 3. Case Studies: Comparative Analysis

In this section, we outline a unique distinct case study (Annual Airline Gala), each with unique WHS considerations. We have tasked an AIc to identify all potential WHS hazards within these scenarios, assess the associated risks, and suggest practical control measures that comply with relevant jurisdictional regulations. The objective is to determine the AIc's utility in recognising and responding to complex WHS issues by adhering to regulatory standards.

Figure 2 is assumed to be an imaginative and highly stylised concept for an event venue suggested and provided by a stakeholder, as three options (i.e., event producer and relevant contractors), themed around aviation and specifically tailored for an airline's annual celebration. It shows a commercial airplane dissected into various sections, with parts of the aircraft repurposed as functional areas for the event. The front section of the plane is transformed into a staging area, complete with lighting and a podium, likely intended for speakers and presentations of national awards. Rows of seats, which resemble airline seating, are neatly arranged in front of the stage for the guests, mimicking the seating arrangement of an airplane but with more space and luxury. In the middle and rear sections of the plane, dining areas are created with round tables set for a fine dining experience, suggesting that a meal may be part of the event's proceedings. These tables are elegantly arranged, complete with table settings and centrepieces, which indicate a formal event. The aircraft's engines serve as unique standing structures on the sides, and the entire setup is placed on what appears to be a blueprint or schematic of an airport or an aircraft layout. This schematic includes markings for various parts of a typical airport, such as gates and jetways, integrating the event space within the theme.

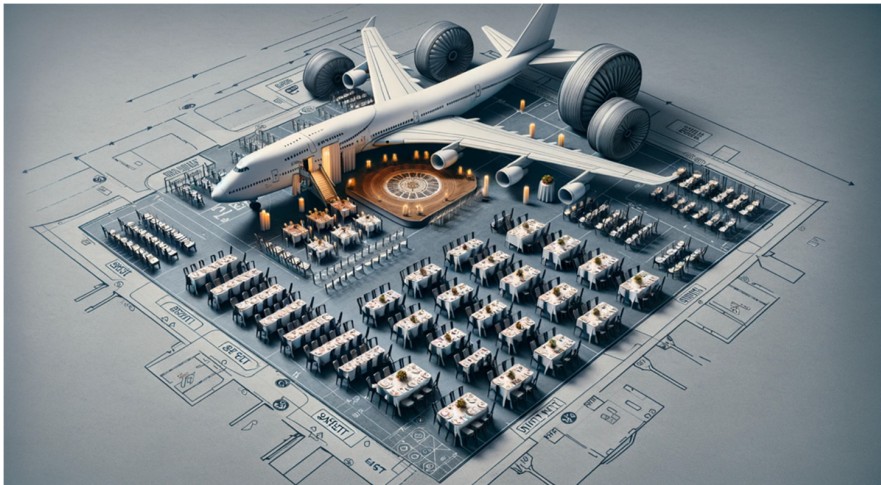

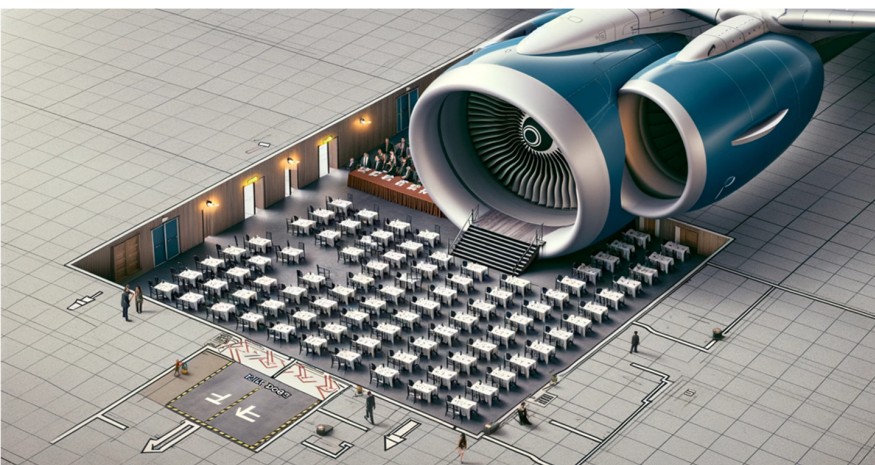

**Figure 2.** *Cont.*

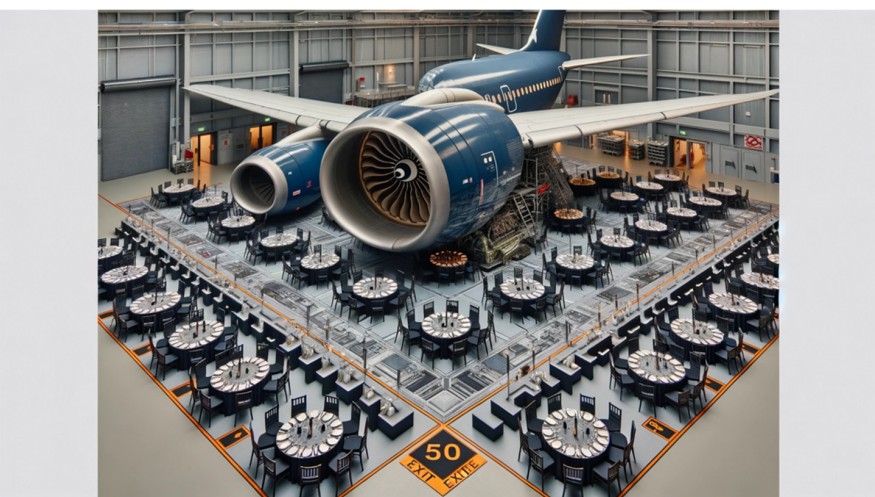

**Figure 2.** The three floor plans of Annual Airline Gala, assuming suggested by stakeholder designer (AI-generated).

The following assumptions have been taken and defined for AIc:

- The event's execution requires the structuring of three distinct phases: setup (bump in), the event duration (live), and dismantling (bump out).
- The venue for the event is situated in New South Wales (NSW), Australia.
- The event is designed to be held indoors.
- There is no existing historical precedent or detailed descriptions available for an event of this nature.

Next, we asked AIc to carry out compressive risk management within three trials, which lastly updated its information in January 2022. The results of risk management using a simple risk matrix are created in less than a minute and presented in Table 1 (the risk matrix is provided in Appendix A). The risk assessment for the Annual Airline Gala in NSW, Australia, has identified potential hazards ranging from physical (like trips and falls, fire hazards), health-related (food poisoning, disease spread), safety concerns (unauthorised access, emergency egress), and environmental issues (spills, waste, air quality). Control measures are comprehensive, aligning with NSW regulations and involve detailed planning, staff training, infrastructure adjustments, and monitoring to mitigate risks to an acceptable level, with a clear delineation of responsibilities among various designated risk owners. Following that, we performed our risk management for Annual Airline Gala, which was fully performed by subject matter experts (SMEs) and is presented in Table 2. It should be added that risk assessors possess relevant educational knowledge in safety as well as substantial industrial expertise. Additionally, the procedure for developing the risk matrix is detailed in the second part of the Appendix A.

**Table 1.** The risk management of Annual Airline Gala based on the AIc.

| Hazard Category | Specific Hazard | Likelihood | Consequence | Risk Level | Control Measures | Risk Owner |
|---|---|---|---|---|---|---|
| Physical | Trips and falls | Likely | Minor to moderate | Medium | • Install high-quality cable protectors compliant with Australian Standards for trip hazard prevention.<br>• Conduct a pre-event inspection to identify and rectify potential trip hazards.<br>• Ensure pathways are always kept clear, with clear signage indicating walkways.<br>• Provide sufficient lighting throughout the venue, especially in walkways and staircases. | Facilities Manager |
| Physical | Fire hazards | Possible | Severe | High | • Strict adherence to NSW Fire and Rescue regulations regarding fire safety requirements.<br>• Regular inspection and maintenance of all fire safety equipment, including extinguishers, hoses, and alarms.<br>• Detailed fire safety training for all event staff, with designated fire wardens.<br>• Clear and well-marked fire exits, with unobstructed pathways leading to them.<br>• Conduct fire drills prior to the event. | Safety Officer |
| Physical | Overcrowding | Possible | Moderate to severe | High | • Calculate and enforce maximum occupancy based on space size as per NSW regulations.<br>• Develop a comprehensive crowd management strategy, which includes defined entry and exit flow management.<br>• Train all staff in crowd control measures and emergency evacuation procedures.<br>• Have dedicated personnel to monitor crowd density and behaviour throughout the event.<br>• Use barriers and signage to guide attendee movement. | Event Coordinator |

**Table 1.** *Cont.*

| Hazard Category | Specific Hazard | Likelihood | Consequence | Risk Level | Control Measures | Risk Owner |
|---|---|---|---|---|---|---|
| Health | Food poisoning | Possible | Minor to severe | Medium | • Strict compliance with FSANZ food safety standards.<br>• Ensure all catering staff hold current NSW Food Authority Food Safety Supervisor qualifications.<br>• Regular food safety audits of all food handling and preparation areas.<br>• Temperature controls for all food storage, preparation, and serving areas.<br>• Clear labelling and separation of allergen-containing foods. | Catering Manager |
| Health | Spread of diseases | Possible | Minor to severe | Medium | • Follow current NSW Health guidelines on communicable diseases, including vaccination recommendations.<br>• Install multiple hand sanitation stations across the venue.<br>• Implement a policy for sick attendees or staff, advising them to stay home.<br>• Have a medical team on standby for any health-related incidents.<br>• Maintain good ventilation systems to reduce the risk of airborne diseases. | Health and Safety Officer |
| Safety | Unauthorised access | Possible | Moderate to severe | Medium | • Implement an access control system in line with NSW WHS Regulation 2017.<br>• Security personnel present at all event entrances and critical areas, trained in line with SLED requirements.<br>• Use of identification badges for staff and attendees.<br>• Have a clear protocol for dealing with unauthorised or suspicious individuals.<br>• Regular security briefings and updates throughout the event. | Security Manager |
| Safety | Emergency egress | Possible | Severe | High | • Ensure all exit signs and emergency egress pathways are up to Australian Building Code standards.<br>• Arrange for clear and practiced evacuation procedures to be in place, familiar to all staff.<br>• Always keep all exits and emergency egress paths clear from obstructions.<br>• Conduct regular evacuation drills in the lead-up to the event.<br>• Have designated personnel to guide attendees in case of emergency. | Compliance Officer |

**Table 1.** *Cont.*

| Hazard Category | Specific Hazard | Likelihood | Consequence | Risk Level | Control Measures | Risk Owner |
|---|---|---|---|---|---|---|
| Environmental | Spills and waste | Possible | Minor to moderate | Medium | • Develop a waste management plan in accordance with NSW EPA regulations.<br>• Provide adequate spill kits and train staff on how to use them properly.<br>• Designate areas for waste collection and ensure these are managed throughout the event.<br>• Have an environmental impact assessment completed, if necessary.<br>• Ensure proper disposal of hazardous materials, including any fuel or chemicals. | Environmental Manager |
| Environmental | Air quality | Possible | Minor to moderate | Medium | • Monitor indoor air quality, ensuring it meets SafeWork NSW guidelines.<br>• Implement a non-smoking policy within the event space.<br>• Utilise air purification systems, if necessary, especially in areas of high congestion.<br>• Prohibit the running of any aircraft or vehicle engines inside enclosed spaces unless adequate extraction and ventilation systems are in place.<br>• Regular checks and maintenance of HVAC systems. | Facilities Manager |
| General | All categories | - | - | - | • - Conduct a thorough WHS risk assessment for each phase of the event (setup, live, dismantling).<br>• - Provide comprehensive WHS training for all staff, tailored to their specific roles and responsibilities.<br>• - Establish a WHS committee comprising members from different departments for the event duration.<br>• - Ensure availability of qualified first aiders and a fully equipped first aid station at the event.<br>• - Have emergency contact numbers readily available. | WHS Committee |

**Table 2.** The risk management of Annual Airline Gala based on the SMEs (i.e., risk analysts).

| Risk Source | Hazard | Likelihood | Consequence | Inherent Risk | Risk Controls | Residual Risk | Risk Owner |
|---|---|---|---|---|---|---|---|
| Working with electrical equipment (setup of audiovisual, amperes, and lux, as well as powered sets). | Electric shock (arc) to a worker while handling equipment during bump-in and bump-out | 3, (Possible—The risk might occur) | 3, (Moderate Injury or Illness requiring hospitalisation via ambulance) | 9, (Medium injury or Illness requiring hospitalisation) | Power mains to be turned off (de-energised) and physically unplugged while the electrical grid and appliances are being plugged/unplugged. A. Connect appliances to truss. B. Plug cords along truss, or as required. C. Plug cords in designated RCD board or controller with RCDs. D. Check that appliances and board switches are in OFF position. E. Connect to mains power. F. Turn mains power on and turn other switches on as required. Load-out to follow the reverse of the above sequence (e.g., F to A) with power feed OFF and disconnected while technicians remove equipment. Mains and portable distribution boards to be RCD-protected and Earth-protected. <br> Crew to test RCDs daily and confirm functionality. Reset with buddy-check immediately after test. <br> No hard wiring unless completed by a licensed electrician. Leads/cords to be uncoiled while energised. Leads not to be connected to other leads for extension, thus potentially increasing RCD trip times. <br> Leads and appliance boards current AS 3760 test, test tags. RCD to bear current RCD test tags. Equipment that does not bear current test tag to be removed from service. <br> Access to power mains and distribution boards to be isolated by keeping doors shut and locked as much as practicable. <br> Production technicians to assess power usage ensuring there is no overload. | 3 (Low) | "Name of Contractors" * |

**Table 2.** *Cont.*

| Risk Source | Hazard | Likelihood | Consequence | Inherent Risk | Risk Controls | Residual Risk | Risk Owner |
|---|---|---|---|---|---|---|---|
| Event management structure and hierarchy not matched with with suggested emergency procedures and systems. | Failure of emergency procedures—failure to respond and evacuate on time due to confusion and unclear hierarchy | 3, (Possible—The risk might occur) | 2, (Minor injury or temporary ill health requiring treatment by medical practitioner) | 6, (Medium) | "Name of Contractors" emergency procedures to be made available to all key event positions/ Individuals. "Name of Contractors" representative to instruct key event personnel on emergency response procedures. of the venue. Occupant capacity to be kept under the maximum rating of the premises. "Name of Contractors" (Emergency Control Organisation) to be interfaced with event management positions. Emergency communications and hierarchy to be established and confirmed between. "Name of Contractors" management and "Name of Contractors" ECO members—in evacuation decision venue ECO take precedence over "Name of Contractors" management. "Name of Contractors" consult and agree upon communication protocol if an emergency occurs during any stage of the event. Fire Fighting Equipment (FFE) to be checked by respective venue representative. Clearances of equipment from emergency exits, evacuation paths, and any potential obstructions of sightlines to emergency exits to be checked and verified by "Name of Contractors" representative. First aid capabilities to be in place at all stages of the event (bump-in, event mode, bumpout) - EMS Event Medical engaged by "Name of Contractors" for first aid/medical services. | 3 (Low) | "Name of Contractors" |

| Risk Source | Hazard | Likelihood | Consequence | Inherent Risk | Risk Controls | Residual Risk | Risk Owner |
|---|---|---|---|---|---|---|---|
| Manual tasks—handling equipment. | Musculoskeletal disorders suffered by workers (MSDs, sprains and strains) | 3, (Possible—The risk might occur) | 2, (Minor injury or temporary ill health requiring treatment by medical practitioner) | 6, (Medium) | Contractors are required to provide mechanical aids such as trolleys or dollies for transporting large objects or equipment and must promote their usage. For large items that cannot be accommodated by mechanical aids, team-lift techniques should be employed. Such lifts should involve pairs or groups of workers with comparable strength and stature, coordinated with commands such as "1-2-3-Lift" and "1-2-3-Lower," typically directed by a supervisor. All items should have their weight clearly indicated to inform workers of the load before handling. Areas on items intended for hand grips should be marked with high-visibility tape or other noticeable markings. Items exceeding 15 kg in weight or larger than 72 cm in any dimension should only be lifted using mechanical aids or through a team-lift approach. Workers should not be compelled to lift items that exceed their personal handling capacity. Workers should engage in appropriate warm-up exercises prior to lifting any large or heavy items and must receive training in proper team-lifting techniques. A qualified supervisor should oversee the lifting process to ensure safety. During these operations, workers must adhere to specific safe work procedures (SWPs) and wear the correct personal protective equipment (PPE) suitable for the task at hand. | 3, (Low) | All Contractors |

**Table 2.** *Cont.*

| Risk Source | Hazard | Likelihood | Consequence | Inherent Risk | Risk Controls | Residual Risk | Risk Owner |
|---|---|---|---|---|---|---|---|
| Leads/cords and other equipment or structures. | Trip and fall | 4, (Likely—The will probably occur) | 3, (Moderate injury or Illness requiring hospitalisation via ambulance) | 12, (High) | Stands' floor platforms must include ramps that adhere to a recommended gradient of 1:14. Ramps should feature a colour that contrasts with the existing floor and platform surfaces to enhance visibility for all users. The stands' flat surfaces must meet a minimum slip resistance rating of P5/R12. Cleaning staff must be readily available to address and clean spills immediately, with an established protocol for reporting spills without delay. Cords or leads should be suspended overhead, maintaining a clearance of at least 2.5 m from the floor to prevent trip hazards. If suspending cords overhead is not feasible, cords must be routed along the base of walls or steps, ensuring they do not cross paths used frequently by pedestrians or in critical areas of thoroughfare. Should the above options be unviable, cords must be securely fixed to the floor with high visibility, contrasting adhesive tape to warn of potential tripping hazards. Furniture should be strategically arranged throughout the area to maintain clear pathways, particularly in the vicinity of stairs. Furniture and equipment must not block emergency exits or evacuation routes. The Safety Officer, in conjunction with the "Name of Contractor," must inspect these arrangements. Steps within sets must have nosings treated with a contrasting colour to ensure clear visibility and to delineate edges. The "Name of Contractor," along with Site-Specific Events and in consultation with the Safety Officer, will evaluate areas designated for carpet installation. Collaboration with AU Carpet is required to ensure the following: a. Suitable protective coverings and hard protection are in place before laying the carpet. b. The carpet is installed without any creases and is securely fixed to prevent tripping and falling incidents. | 8, (Medium) | "Name of Contractors" |

**Table 2.** *Cont.*

| Risk Source | Hazard | Likelihood | Consequence | Inherent Risk | Risk Controls | Residual Risk | Risk Owner |
|---|---|---|---|---|---|---|---|
| Common combustibles—furniture, signage—in proximity to event electrical equipment. | Fire and fire-related injuries such as burns, toxic smoke inhalation. | 4, (Likely—The will probably occur) | 3, (Moderate injury or Illness requiring hospitalisation via ambulance) | 12, (High) | All production equipment, particularly lighting and audio-visual (Lx, AV) known to generate heat, must be installed by qualified technicians at a safe distance from drapes and any flammable materials to mitigate fire risk. The contractor's designated manager, alongside the Safety Officer, must regularly verify the operational status of fire detection and suppression systems, as well as the readiness of fire extinguishers, ensuring inspection tags are current. All electrical equipment must adhere to the specific risk controls outlined in the provided risk register document. Production and lighting (Lx) technicians are to prioritise the use of low heat-emitting LED fixtures. If the use of high-intensity discharge lamps such as "sharpie" lights is unavoidable, the Lx contractor must ensure these are positioned well away from any combustible materials. The contractor's manager is responsible for routinely inspecting emergency exits to ensure that no equipment or materials are obstructing them, thus maintaining clear evacuation paths at all times. The contractor is required to adhere to the established firefighting equipment plan, ensuring that the location, type, and maintenance of firefighting resources meet or exceed the detailed specifications. | 8, (Medium) | "Name of Contractors" |

**Table 2.** *Cont.*

| Risk Source | Hazard | Likelihood | Consequence | Inherent Risk | Risk Controls | Residual Risk | Risk Owner |
|---|---|---|---|---|---|---|---|
| Equipment attached overhead and rigging. | Collapse of overhead equipment causing injuries (errors or omissions, wear and tear, overload). | 3, (Possible—The risk might occur) | 3, (Moderate injury or illness requiring hospitalisation via ambulance) | 9, (High) | Contractors must utilise only the venue-approved rigging nods to ensure safety and compliance with standards. Overhead production equipment must be secured with original manufacturer's fastening gear, such as clamps and frames, or attachment methods sanctioned by a certified rigger with event or film industry expertise. Safety measures must include the attachment of ropes (specifically flexible steel wire ropes) or chains connecting the overhead gear to the venue's designated ceiling anchor points. It is essential for contractors to engage in dialogue to validate that the chosen ceiling points are structurally capable of bearing the loads, especially for heavy custom elements. All rigging hardware must display a clear WLL (working load limit), and the setup must adhere to a safety factor of 10. Only a certified professional with an advanced rigging license and relevant field experience should verify the rigging's adequacy. Contractors should consider implementing static lines for truss systems above audience areas and performance stages. Exclusion zones must be enforced to prevent personnel from standing beneath trusses during lifting operations. Cable looms require securement with appropriately rated slings and shackles to prevent displacement. Ensure the locking pins of critical shackles are properly secured (moused) to prevent loosening. Use steel rope slings in proximity to areas where special effects are deployed for enhanced safety. | 1, (Very Low) | "Name of Contractors" |

**Table 2.** *Cont.*

| Risk Source | Hazard | Likelihood | Consequence | Inherent Risk | Risk Controls | Residual Risk | Risk Owner |
|---|---|---|---|---|---|---|---|
| Lack of site safety inductions and briefings. | Workers unaware of, or unfamiliar with, work environment, safety arrangements and rules. Incidents and injuries due to breaches of site/venue safety rules. | 3, (Possible—The risk might occur) | 3, (Moderate injury or illness requiring hospitalisation via ambulance) | 9, (High) | The Safety Officer, in conjunction with a representative from the contracting company, will conduct safety inductions for all workers on site. Induction topics for workers include, but are not limited to: Protocols for safety reporting and consultation; The contractor's safety regulations; Site-wide compulsory personal protective equipment (PPE) and additional PPE recommended by the contractor; Security measures and access control, including restricted areas; Processes for safety issues reporting, consultation, and resolution; Details about on-site amenities and facilities; Emergency response procedures, such as evacuation routes, the locations of assembly points, the roles of emergency wardens, and contact information for first aid officers; Specific venue constraints and areas requiring special access permits. Workers must acknowledge their induction by signing forms or completing an online induction process, as required. It is the responsibility of the contracting company and the Safety Officer to verify that all workers have completed the induction process before commencing work. | 1, (Very Low) | "Name of Contractors" |
| Plant operation—forklift truck (FLT). | Worker hit or crushed by plant. | 3, (Possible—The risk might occur) | 4, High, (one or more fatalities or permanent disability/ill health to one or more persons) | 12, (High) | All plant operators must possess a valid high-risk work license with a photo ID, an expiry date, and an LF classification. The Safety Officer is responsible for conducting periodic checks of these licenses. Areas of plant operation should be clearly marked and separated from pedestrian zones as much as feasible to ensure the safety of workers on foot. The safe working load (SWL) for all plant equipment must be rigorously observed. Forklift trucks (FLTs) should be used in compliance with the Australian Standard AS 2359 Part 2, which covers the operations of powered industrial trucks. When in use, forklifts must have their headlights on, and operators are required to sound the horn when approaching blind corners to alert nearby workers. On-site storage of FLT gas bottles should be minimal, and they must be stored upright in secured cages with the valves at the top. Forklifts must be operated at a safe speed, equivalent to a walking pace, to prevent accidents and ensure pedestrian safety. All personnel involved in the setup (bump-in) and dismantling (bump-out) stages must wear high-visibility clothing that meets the Australian Standard AS 4602 for both daytime and night-time use. | 3, (Low) | "Name of Contractors" |

**Table 2.** *Cont.*

| Risk Source | Hazard | Likelihood | Consequence | Inherent Risk | Risk Controls | Residual Risk | Risk Owner |
|---|---|---|---|---|---|---|---|
| Plant operations—MEWP. | Worker falling off plant e.g., MEWP | 3, (Possible—The risk might occur) | 4, (Serious injury or illness requiring immediate hospital admission via ambulance) | 12, (High) | MEWPs should be operated in such a manner that eliminates the need for operators to overreach, reducing the risk of imbalance or falls. Crew chiefs are responsible for ensuring compliance, with oversight from the Safety Officer.<br>Operators and any passengers of MEWPs must wear full-body harnesses, which are to be securely attached via a fall arrest system (including a lanyard and karabiners) to the designated anchor points on the equipment (such as on a boom lift). Additionally, they are required to wear rigging helmets to protect against potential head injuries from impacts with overhead structures.<br>Tools should be tethered to the operator's hands or to a secure point to prevent them from being dropped from height, which could pose a risk to people below or result in property damage.<br>Personnel are prohibited from climbing on the handrails of the MEWP basket or exiting the basket at height, ensuring they remain within the confines of the safety rails.<br>The MEWP should be used in a low-speed setting, often referred to as "turtle mode" or creep mode, to maintain precise control during operations.<br>Both the operator and any passengers in the MEWP basket must use full-body harnesses with lanyards attached to the basket, ensuring they are protected in case of a fall.<br>The operation of MEWPs must comply with the requirements of the Australian Standard AS 2550.10, which governs safe use practices for this type of machinery. | 4, (Low) | "Name of Contractors" |

**Table 2.** *Cont.*

| Risk Source | Hazard | Likelihood | Consequence | Inherent Risk | Risk Controls | Residual Risk | Risk Owner |
|---|---|---|---|---|---|---|---|
| Plant operations—MEWP. | Person crushed by MEWP | 3, (Possible—The risk might occur) | 4, (Serious injury or illness requiring immediate hospital admission via ambulance) | 12, (High) | Secure a MEWP from a trusted supplier for use in high-level rigging tasks and for facilitating the assembly and disassembly of seating stands.<br>Schedule MEWP operations for times when the fewest number of workers are present on-site to minimise risk, and ensure operation is either overseen by a qualified ground spotter or conducted within a designated, clear area.<br>For extended use of MEWPs, prioritise electrically powered units to avoid emissions. If a diesel-powered unit must be used, turn off the engine when the platform is stationary to prevent unnecessary exhaust fumes.<br>Operate MEWPs only on stable, level terrain capable of supporting the machine's full weight to prevent tipping or collapse.<br>Ensure all MEWP operators hold the appropriate licences, including a "WP" High Risk Work Licence for those operating equipment with a reach exceeding 11 m, or a competency card from the Australian MEWP Operators Association for scissor lifts. Implement sporadic licence verifications by the Safety Officer and the responsible contractors.<br>Enforce a strict no-access zone under the MEWP booms and within an 8 m radius of the machine to safeguard against potential falls or drops.<br>Turn off diesel-powered MEWPs when not actively in use to prevent the accumulation of harmful fumes, and ensure electric MEWPs are charged in areas with adequate ventilation to mitigate any risk of fire or fume accumulation overnight. | 6, (Medium) | "Name of Contractors" |

**Table 2.** *Cont.*

| Risk Source | Hazard | Likelihood | Consequence | Inherent Risk | Risk Controls | Residual Risk | Risk Owner |
|---|---|---|---|---|---|---|---|
| Works on ladders. | Fall of worker from ladder and related injury. | 4, (Likely—The risk will probably occur) | 3, (Moderate injury or illness requiring hospitalisation via ambulance) | 12, (High) | Contractors should strategise work to reduce the necessity for working at height, planning tasks to be completed primarily at ground level to minimise the use of ladders. Provide industrial-grade platform A-frame ladders that enable secure, hands-free operations, enhancing safety for tasks that require working at a height. All ladder-related activities must be under strict supervision to ensure adherence to safety protocols. Ensure that ladders are positioned on stable, even surfaces, avoiding placement in high-traffic areas such as doorways and corridors to prevent obstructions and potential accidents. Ladder safety must be a priority: workers should avoid using the top two rungs for standing, overreaching, or carrying objects while climbing or descending. Workers should face the ladder at all times where possible to maintain balance and control. Instruct workers to apply their body weight towards the ladder base when necessary to enhance stability, especially when working on softer grounds or uneven surfaces. Prohibit the use of ladders that are identified as damaged or faulty to prevent accidents. Such ladders must be removed from the work area and clearly marked or tagged to avoid accidental use. Ensure that "Name of Contractors" have effective first aid measures in place and readily accessible throughout the entire duration of the event, including trained first aid personnel and fully stocked first aid kits. | 4, (Low) | "Name of Contractors" |
| Site visitors during works (load-in in particular). | Injury to a "Name of Contractors" representative or "Name of Contractors" staff member due to entry in hazardous work areas. | 2, (Unlikely—The risk is not expected to occur in most circumstances) | 1, (Very low, minor injury or temporary ill health requiring treatment by medical practitioner) | 2, (Low) | Collaboratively, "Name of Contractors" and the event venue should develop and implement a comprehensive access control plan specifically for the load-in and load-out phases to ensure safety and security. Implement a strict policy whereby non-participating visitors are restricted to certain areas unless accompanied directly by a properly inducted and qualified project manager or crew chief to access active work zones. "Name of Contractors" should evaluate the benefits of requiring all individuals present on the site during critical operational times such as load-in and load-out to wear high-visibility clothing to enhance on-site safety through better visibility. "Name of Contractors" should devise a clear and effective procedure for managing visitors who do not comply with safety instructions or site rules, ensuring that there are consequences for non-compliance to maintain a secure and safe environment. | 1, (Very Low) | "Name of Contractors" |

**Table 2.** *Cont.*

| Risk Source | Hazard | Likelihood | Consequence | Inherent Risk | Risk Controls | Residual Risk | Risk Owner |
|---|---|---|---|---|---|---|---|
| Amplified PA sound. | Noise levels levels causing worker hearing deterioration. | 3, (Possible—The risk might occur) | 1, (Minor injury or temporary ill health requiring treatment by medical practitioner) | 3, (Medium) | Audio technicians are tasked with calibrating speaker and amplifier outputs to levels that safeguard against hearing impairment, in line with the New South Wales Code of Practice, which identifies 70 decibels as equivalent to the volume of a loud conversation. "Name of Contractors" are responsible for conducting sound level checks using an objective methodology to ensure that two individuals standing a meter apart can communicate comfortably without the need to raise their voices, even while presentations are ongoing onstage. "Name of Contractors" must establish a system that allows for the immediate reduction of amplified sound should any staff member report auditory discomfort or concern. | 1, (Very Low) | "Name of Contractors" |
| Dark areas—theatrical lighting calling for some reduction of illumination levels in the premises. | Mechanical injury to a worker—trip and fall, walk into an object. | 3, (Possible—The risk might occur) | 3, (Moderate injury or illness requiring hospitalisation via ambulance) | 9, (High) | Ensure consistent illumination along walkways, corridors, and access routes to facilities throughout the duration of the event, giving special attention to areas where lighting may be dimmed or altered for event purposes. Implement additional lighting in locations where temporary changes in elevation occur, such as the installation of stairs or ramps, or where floor gradients change, to prevent missteps and falls. Engage in a thorough walk-through of the event site in collaboration with the Safety Officer, methodically evaluating the lighting conditions and identifying any potential trip and fall hazards. Create a visual record of the event's spaces through detailed photography to assist in safety assessments and provide a reference for future events. | 4, (Low) | "Name of Contractors" |

**Table 2.** *Cont.*

| Risk Source | Hazard | Likelihood | Consequence | Inherent Risk | Risk Controls | Residual Risk | Risk Owner |
|---|---|---|---|---|---|---|---|
| Special effects: Coolamon, Low Fox, Haze, Sparkulars, Streamer Cannons, CO$_2$ Jets. | Catastrophic fire in the context of aircraft A1 fuel fumes dissipating from the A380 wings, cold burns. | 3, (Possible—The risk might occur) | 5, (One or more fatalities or permanent disability/ill health to one or more persons) | 15, (High) | Engage in a thorough and fact-based safety assessment regarding the use of Sparkulars or similar effects in proximity to aircraft, especially considering their compatibility with aviation fuel fumes. Online research indicates a low temperature of discharge, but an in-depth analysis specific to the aviation context is necessary. Consider the adoption of alternative cold special effects that have been validated for use near aircraft by competent technicians. Consult with insurance providers to fully understand the implications and liability of the risk involved. Ensure that CO$_2$ containers are fastened securely and positioned with their valves oriented upwards to prevent any accidental release or mishandling. Designate firing positions with clearly marked safety exclusion zones. These areas should be continuously monitored to ensure they remain clear of unauthorised personnel during the event. Technicians should maintain a clear line of sight to each pyrotechnic firing position to efficiently manage cues and abort ignitions, if necessary, should individuals enter the safety exclusion zones. Carefully evaluate the setup of Streamer Cannons, including the angles and force of discharge, to predict the trajectory and fallout area of streamers or other effects to ensure audience safety and prevent any contact or injury. Replace any effects that involve open flames, such as the Coolamon walk/effect, with non-flame alternatives to eliminate fire hazards, particularly in environments where there's a risk of fire or in compliance with fire safety regulations. | 4, (Low) | "Name of Contractors" |
| The venue location is a purpose built premises—an aircraft maintenance workshop not inherently intended to accommodate large number of occupants. | Injury or illness due to inability to evacuate timely | 3, (Possible—The risk might occur) | 4, (Serious injury or illness requiring immediate hospital admission via ambulance) | 12, (High) | Ensure that all emergency exits, particularly those located to the East of Hangar 96, remain clear of obstructions at all times, with exit doors being readily operable to facilitate quick evacuation. Conduct a detailed inspection of the eastern shutter door to confirm its reliability for use as an emergency exit. Note that the egress capacity without the use of this shutter is calculated at 131 persons per minute. Utilising the shutter increases the total egress capacity to 508 persons per minute, which is critical in achieving the desired evacuation time frame. Task the Safety Officer with performing a comprehensive check of all emergency exits and the roller shutter/door designated for emergency use to confirm their operational status and accessibility. This inspection should be documented with photographic evidence to ensure that evacuation routes are fully functional and meet safety standards. | 4, (Low) | "Name of Contractors" |

**Table 2.** *Cont.*

| Risk Source | Hazard | Likelihood | Consequence | Inherent Risk | Risk Controls | Residual Risk | Risk Owner |
|---|---|---|---|---|---|---|---|
| Event freight vehicle operations (event delivery dependant on substantial amount of trucking of equipment). | Worker struck by a vehicle. Event freight vehicle involved in a motor vehicle accident (MVA) such as collision with another vehicle or plant item. | 3, (Possible—The risk might occur) | 4, (Serious injury or illness requiring immediate hospital admission via ambulance) | 12, (High) | Collaborate with "Name of Contractors" and site-specific events to develop a comprehensive vehicle movement plan. The plan should comply with all airport vehicle restrictions, including limitations on size, mass, and height, and configuration. It should prioritise the selection of vehicles that can be easily and safely navigated through the event's access routes. Consideration should be given to designing vehicle paths that reduce or eliminate the need for complex manoeuvres such as reversing, sharp U-turns, or 3-point turns. Ensure proactive communication of the vehicle movement plan to all involved contractors well before the event setup to ensure seamless operations. Engage a Safety Officer to provide vehicle spotting assistance as needed to ensure the safe movement of vehicles around the site. Mandate that every person present on site during the setup (bump-in) and breakdown (bump-out) periods wear high-visibility workwear to enhance on-site safety. | 4, (Low) | "Name of Contractors" |
| Reliance on PPE for capture of exposures to residual risks from various hazards. | Poor PPE discipline leading to injury. Lack of PPE or lack of adequate wear of PPE based on the hazard. | 3, (Possible—The risk might occur) | 4, (Serious injury or illness requiring immediate hospital admission via ambulance) | 12, (High) | "Name of Contractors" to enforce the wearing of high-visibility workwear as mandatory personal protective equipment (PPE) for all personnel during setup (bump-in) and breakdown (bump-out) phases. In a pre-event contractor briefing, "Name of Contractors" to advise on additional PPE measures in light of the presence of a resident falcon known for aggressive behaviour. Recommendations include eye protection and safety helmets, with a strong emphasis on protection for mobile elevating work platform (MEWP) operators. The Safety Officer should detail these additional PPE recommendations, as outlined in the document "SIM SIN 220826 Mandatory Personal Protection Equipment," during safety inductions, specifically addressing the risks associated with the carnivorous bird. For MEWP operations: A. Mandate the wearing of safety helmets by all individuals in the vicinity of MEWP operations. B. Clearly demarcate MEWP operational areas and enforce the wearing of safety helmets within these zones. Ensure that all contracting firms have robust supervision and management practices in place to provide PPE as required by Safe Work Procedures (SWPs) and Safe Work Method Statements (SWMS), potentially including eye protection, gloves, protective footwear, and hearing protection. Obligate contractors to actively monitor and manage their workforce's compliance with PPE requirements and have a supply of spare PPE available to address non-compliance immediately. | 1, (Very Low) | "Name of Contractors" |

**Table 2.** *Cont.*

| Risk Source | Hazard | Likelihood | Consequence | Inherent Risk | Risk Controls | Residual Risk | Risk Owner |
|---|---|---|---|---|---|---|---|
| Gas containers on site (FLT and catering). | Gas leak and fire or explosion. Gas containers compromised by plant of vehicles moving nearby and striking them. | 3, (Possible—The risk might occur) | 5, (One or more fatalities or permanent disability/ill health to one or more persons) | 15, (High) | Gas suppliers must verify that all gas containers used are within their 10-year pressure test certification period before selection for use. Limit the amount of liquefied petroleum gas (LPG) within the hangar to the minimum necessary for event-specific food preparation. All fork lift truck (FLT) LPG containers should be stored in a specially designed gas cage equipped with appropriate safety markings, including clear "No Open Flame" signs. The gas storage cage must be situated in a location that is well away from plant operations and vehicular traffic. This cage must remain closed and securely locked at all times to prevent unauthorised access. "Name of Contractors" must take responsibility for ensuring: The use of cooking appliances rated safe for indoor use within the hangar. The installation of blow-back valves on all gas connections to prevent reverse gas flow. The secure fastening of all gas containers in an upright position using metal chains, ensuring they are anchored to sturdy objects and valves are oriented upwards. The engagement of a certified gas fitter to manage fittings for large-capacity gas containers and to guarantee that compliance certification plates are visible. The performance of soap bubble tests and thorough visual inspections to confirm the absence of gas leaks. The placement of container valves in positions that are readily accessible for quick shutdown in the event of a leak. "Name of Contractors" must procure an up-to-date Safety Data Sheet (SDS) for the types of gas in use on the premises and maintain a hard copy readily accessible on site. The Safety Officer is tasked with the ongoing monitoring of adherence to these guidelines and must take immediate action to correct any deviations from established safety practices. "Name of Contractors" in conjunction with the Safety Officer, must ensure that all necessary firefighting equipment is not only available but also clearly marked with proper signage for rapid identification and access. | 4, (Low) | "Name of Contractors" |

**Table 2.** *Cont.*

| Risk Source | Hazard | Likelihood | Consequence | Inherent Risk | Risk Controls | Residual Risk | Risk Owner |
|---|---|---|---|---|---|---|---|
| Glass and porcelain used during the gala dinner. | Accidental break of glass or ceramics contributing to incisions or lacerations. Shrapnel affecting sensitive aircraft tyres post event if not cleaned | 3, (Possible—The risk might occur) | 4, (Serious injury or illness requiring immediate hospital admission via ambulance) | 12, (High) | "Name of Contractors" should evaluate the use of modern Perspex vessels for serving purposes to ensure that they meet the event's presentation standards. If these vessels are determined to be suitable without compromising presentation quality, they should be utilised. "Name of Contractors" must secure the services of a professional event cleaning company. It should be ensured that there are sufficient numbers of trained cleaners equipped with the necessary tools on standby to promptly and efficiently address any incidents involving broken glass or porcelain. "Name of Contractors" is responsible for organising a comprehensive cleaning operation for the entire event space following the conclusion of the event. This is particularly important in cases where breakage of materials like glass or porcelain has occurred. After the initial cleaning is completed, "Name of Contractors" must arrange for an extensive floor cleaning using a ride-on cleaning machine. This should involve a wet or chemical vacuum sweep to ensure thorough cleanliness of the hangar floor after the event. | 1, (Very Low) | "Name of Contractors" |
| Temporary Demountable Structures (TDS—3 m x 3 m tent, registration booths) and generator placed air side. | Wind forces causing displacement, tip-over or ultimate collapse of a TDS. Generator placement inadequate and presenting a fire risk. | 3, (Possible—The risk might occur) | 4, (Serious injury or illness requiring immediate hospital admission via ambulance) | 12, (High) | "Name of Contractors" has engaged a professional marquee/tent supplier, which is expected to be validated with a Design Certification from a Certified Practicing Structural Engineer. Furthermore, "Name of Contractors" will ensure that additional ballast or kentledge is secured for the structure's stability. "Name of Contractors" will require all suppliers and builders of temporary demountable structures (TDS) to provide assessments by a competent certified practicing structural engineer for structures exceeding 2 m in height and width. For smaller installations, such as registration kiosks, a qualified person (e.g., a builder) must certify the bracing methods and kentledge necessary to maintain structural integrity in winds reaching a minimum of 65 km/h. "Name of Contractors" will diligently monitor weather updates from the Bureau of Meteorology/MetEye, checking twice daily during setup (bump-in) and hourly during the event for any severe weather alerts and wind advisories. Contingency plans will be in place to relocate registration processes inside the hangar and to execute shelter-in-place protocols if severe weather conditions arise. TDS installers are required to provide written confirmation that all structures have been constructed in accordance with the stipulated specifications and are safe for use prior to them being inhabited or utilised by guests. | 4, (Low) | "Name of Contractors" |

| Risk Source | Hazard | Likelihood | Consequence | Inherent Risk | Risk Controls | Residual Risk | Risk Owner |
|---|---|---|---|---|---|---|---|
| Residual risks with potential of an accident and injury on site. Potentially deficient first aid capabilities. | Deterioration of injured worker condition due to inability to render first aid assistance on time. | 3, (Possible—The risk might occur) | 3, (Minor injury or temporary ill health requiring treatment by medical practitioner) | 6, (Medium) | First aid-certified individuals will be present throughout the event duration, ensuring at least one, but preferably two during key times such as setup (bump-in), including roles such as the Safety Officer and security staff. The first aid station will be clearly marked, easily accessible, and its supplies regularly inspected and confirmed to be complete and adequate. During toolbox talks and safety inductions, all workers will be informed about the location of the first aid kit, identify the designated first aid responders, and understand the procedures for reporting incidents and injuries. Appointed first aiders will maintain a log of all first aid interventions in an injury register, ensuring a record of treatments is kept for future reference and reporting requirements. "Name of Contractors" will provide comprehensive guidelines on reporting procedures and the necessary steps for escalation in the event of serious injuries or incidents requiring medical intervention, such as ambulance services or incidents that must be reported to SafeWork NSW. "Name of Contractors" will have arrangements with EMS Event Medical to provide professional medical support during both rehearsals and the actual event to ensure immediate response capabilities for any medical emergencies. | 1, (Very Low) | "Name of Contractors" |

**Table 2.** *Cont.*

| Risk Source | Hazard | Likelihood | Consequence | Inherent Risk | Risk Controls | Residual Risk | Risk Owner |
|---|---|---|---|---|---|---|---|
| Various work groups involved with physically and mentally demanding work tasks. | Worker fatigue, dehydration, exhaustion leading to accident and injury. | 3, (Possible—The risk might occur) | 3, (Moderate injury or illness requiring hospitalisation via ambulance) | 9, (High) | "Name of Contractors" has partnered with select suppliers renowned for their specialist services and for providing teams with seasoned event professionals.<br>"Name of Contractors" will ensure the availability of drinking water, coordinating with the responsible parties to provide access to tap water and disposable cups, or advising crew members to bring their own refillable containers.<br>Contractor crew chiefs or supervisors are responsible for assessing the skill levels of the workforce, matching new or less experienced workers with those who have more event-specific expertise to foster skill development and work efficiency.<br>Supervisors will strategically assign tasks to workers to maintain an even workload distribution, ensuring that tasks are allocated to optimise teamwork and productivity among the crew.<br>It is the role of the crew chiefs and supervisors to be vigilant for any signs of worker fatigue—such as irritability, lack of focus, or forgetfulness—and to intervene promptly by offering regular, short breaks to maintain high levels of work safety and quality.<br>Contractors in collaboration with "Name of Contractors" are to put in place stringent cross-checking procedures for critical tasks, requiring at least two workers and a supervisor to confirm the completion and accuracy of the task before proceeding to subsequent tasks. | 1, (Very Low) | "Name of Contractors" |

**Table 2.** *Cont.*

| Risk Source | Hazard | Likelihood | Consequence | Inherent Risk | Risk Controls | Residual Risk | Risk Owner |
|---|---|---|---|---|---|---|---|
| Presentation/ performance stage—live/ unsecured edges. | Fall of worker, presenter or artist from stage deck and associated injury. | 3, (Possible—The risk might occur) | 3, (Moderate injury or illness requiring hospitalisation via ambulance) | 9, (High) | The construction of the stage is to adhere strictly to the system specifications, including the secure connection of boards and appropriate bracing. The stage's structural capacity must be certified to support a minimum of 5 kN per square meter and 3.6 kN per 5 square centimetres.<br>The stage design should feature a white surface with contrasting black for the flooring and stairs to enhance visual differentiation. The number of workers permitted on stage areas without installed handrails should be kept to a strict minimum to ensure safety.<br>The staging contractor is required to clearly delineate the edges of stage stairs using a conspicuously contrasting tape (white colour is the preferred choice for visibility) to demarcate any elevation changes such as steps, ramps, and deck platforms, ensuring that the platform dimensions are clearly visible.<br>Contractor supervisors are charged with the rigorous enforcement of these safety measures, expressly prohibiting workers from approaching any unsecured stage edges.<br>Designated areas for presenters and performers should be clearly marked on the stage, and sufficient rehearsal time must be provided to allow for familiarity with the stage layout and to ensure a comfortable performance environment.<br>Lighting should be arranged to avoid direct illumination of presenters and performers from angles that would cause discomfort or hinder performance, particularly avoiding lights directed at their faces from straight-ahead angles. | 4, (Low) | "Name of Contractors" |

**Table 2.** *Cont.*

| Risk Source | Hazard | Likelihood | Consequence | Inherent Risk | Risk Controls | Residual Risk | Risk Owner |
|---|---|---|---|---|---|---|---|
| Reliance on risk controls. | Lack of risk control application and verification leading to worker injury. Emerging hazards not addressed. | 3, (Possible—The risk might occur) | 3, (Moderate injury or illness requiring hospitalisation via ambulance) | 9, (High) | "[Name of Contractors]" are to thoroughly examine this document, formulate a comprehensive plan for the communication of the detailed risk mitigations to all parties involved, and ensure the enactment of the specified controls for which they are responsible to the fullest extent feasible.<br>"[Name of Contractors]" are to appoint specific individuals tasked with the enforcement of the risk mitigations listed, which includes identifying the pertinent subcontractors as noted in this registry. This risk registry should be disseminated amongst all event participants to allow for the opportunity to review, provide input, and be informed about the established controls, as well as to organise their own implementation strategies.<br>"[Name of Contractors]" should utilise this document in its physical form at the event location to confirm that the described mitigations have been applied, recording observations and authenticating compliance alongside each control item.<br>"[Name of Contractors]" must articulate expectations and duties with unequivocal clarity through written communication.<br>"[Name of Contractors]" must assign a team member the role of vigilant overseer for the ongoing operations, tasked with immediate intervention upon the identification of any new or evolving potential hazards. | 4, (Low) | "Name of Contractors" |

**Table 2.** *Cont.*

| Risk Source | Hazard | Likelihood | Consequence | Inherent Risk | Risk Controls | Residual Risk | Risk Owner |
|---|---|---|---|---|---|---|---|
| High "Name of Contractors" reliance on contractor expertise. | Gaps of contractor planning and actual on-the ground compliance leading to incidents and | 3, (Possible—The risk might occur) | 3, (Moderate injury or illness requiring hospitalisation via ambulance) | 9, (High) | "[Name of Contractors]" are tasked with a thorough review of this document and are responsible for disseminating the detailed risk mitigation strategies to all relevant parties.<br>"[Name of Contractors]" shall appoint specific individuals accountable for the enactment of the risk mitigation measures outlined in this document.<br>Contractors are required to provide Safe Work Method Statements (SWMS) and Safe Work Procedures (SWPs), Safety Data Sheets (SDS), and engineering specifications for any temporary structures that can be disassembled, to "[Name of Contractors]". These documents should detail the protocols for safe work execution, including information on the equipment to be utilised, adherence to compliance standards, and the necessary training for their workforce.<br>Contractors are also required to present proof of Workers' Compensation and Public Liability insurance as evidence of their compliance with statutory obligations.<br>"[Name of Contractors]" are responsible for designating a team member whose role will be to collect, scrutinise, and confirm the accuracy of contractor documentation, and to oversee contractor compliance throughout every phase of the event.<br>"[Name of Contractors]" should take into consideration the incorporation of safety performance as a key factor in establishing Key Performance Indicators (KPIs) for contractors. | 4, (Low) | "Name of Contractors" |

**Table 2.** *Cont.*

| Risk Source | Hazard | Likelihood | Consequence | Inherent Risk | Risk Controls | Residual Risk | Risk Owner |
|---|---|---|---|---|---|---|---|
| Inadequate safety communication consultation, coordination and cooperation. | Workplace incident and injury/illness attributable to inadequate stakeholder communication and consultation - errors, omissions or assumptions. | 4, (Likely—The risk will probably occur) | 3, (Moderate injury or illness requiring hospitalisation via ambulance) | 12, (High) | "[Name of Contractors]" should actively engage in communication with all stakeholders involved in the event to ensure a collaborative approach.<br>Obtain written confirmation on critical issues to ensure clarity and accountability.<br>Maximise stakeholder inclusion by utilising group communications, such as emails, to broaden the scope of discussions and input.<br>Direct the attention of contractors and partners towards established timelines for crucial decisions, facilitating adequate preparation time for all parties involved.<br>Ensure that significant decisions are recorded in writing, with a thorough review and confirmation process to verify accuracy and understanding.<br>Foster a culture of communication, consultation, and collaboration among all stakeholders on Work Health and Safety (WHS) matters, in compliance with the WHS Act 2011 and WHS Regulation 2017, aligning with industry best practices.<br>Mandate that all personnel involved with the event receive proper instruction from the Safety Officer regarding the necessary procedures for reporting safety incidents and engaging in safety consultations. | 4, (Low) | "Name of Contractors" |

* The "Name of Contractors" refers to the stakeholders involved in the Airline Gala Dinner, each being responsible for specific risk control measures. This includes the contractors responsible for stage construction, who must ensure structural safety; the lighting and sound engineers, tasked with electrical safety and noise control; the medical personnel on-site for emergency first aid response; and the fire safety professionals prepared to manage and prevent fire-related incidents.

Briefly, when we looked at the differences between the results of the risk management—one created by AI (Table 1) and the other crafted by a human expert (Table 2)—we saw distinct approaches in the level of detail and mitigation strategies. The table born out of AI's capabilities is meticulously organised, listing hazards, potential outcomes, measurable risks, and targeted mitigation steps. It is impressively detailed, featuring actionable measures such as collaborating with emergency responders and maintaining a risk register, which is in line with the best practices of the industry. In contrast, the Table generated by a human expert opts for a wider lens, favouring a narrative style that speaks to broad safety principles such as collective responsibility and constant alertness. While it may not dive into the minutiae as the AI's output does, it champions conceptual strategies to nurture a culture of safety and collective awareness, which are just as crucial as specific measures.

Structurally, both Tables use a columnar format to classify hazards and mitigation measures. The AI's rendition is steeped in formal and technical jargon, tailoring it for industry insiders who operate under rigorous regulatory mandates. Meanwhile, the human expert's rendition employs a more straightforward language, which, while sacrificing some detail, broadens its appeal to a more general audience. The difference in the clarity of information is also telling; the precision of the AI-crafted table cuts through ambiguity, providing a clear-cut guide for implementing safety measures. On the flip side, the human expert's table offers a sketch that invites further discussion and elaboration, making it a prime tool for teaching rather than immediate procedural application. When it comes down to it, the AI's Table shines in providing clear directives and structured risk management ready for action. The human expert's Table, however, excels at laying the groundwork for safety principles and fostering a culture of mindfulness. Within the broader context of a risk management plan, they could play supportive and enriching roles to one another: the AI's output directing specific safety actions and the human expert's insights being pivotal for team discussions and promoting an engaged, safety-first attitude among the workforces. In unison, they offer a dual approach that encapsulates the precision of risk control and the essential cultivation of a conscious approach to workplace safety.

In the following section, we conduct a comprehensive comparison of risk management that includes the analysis of image data, considering multiple criteria.

## 4. Comparative Evaluation

In an era where AI systems like ChatGPT-4 are increasingly employed to supplement human expertise in risk management, it becomes essential to scrutinise their performance critically. This section comprehensively analyses how ChatGPT-4 interprets and responds to risk management tasks across various criteria. We explore the system's effectiveness in identifying hazards, evaluating risks, and proposing control measures while addressing its limitations and the factors influencing its interpretative capabilities. We consider the role of context as a vital component in the application of AI to risk management, acknowledging that the effectiveness of AI is not only in the processing of image data but also in understanding the context within which the risks occur. To evaluate the performance of ChatGPT-4 in these dimensions, we have established a set of criteria (Figure 3) that cover the accuracy, relevance, practicality, response time, comprehensiveness, and contextual understanding of its output.

- **Completeness of Hazard Identification:** This aspect assesses ChatGPT-4's proficiency in identifying all potential hazards in a given image. The evaluation focuses on the model's accuracy in detecting risks without missing any (false negatives) and ensuring that it does not erroneously flag harmless elements as hazardous (false positives). This metric is crucial for establishing the reliability of the AI in hazard detection scenarios.
- **Relevance of Risk Evaluations:** This criterion examines how effectively the AI assesses the severity and likelihood of identified risks. It involves evaluating ChatGPT-4's consistency in risk evaluation across various scenarios, ensuring that its judgments align with the threat levels of different hazards.

- **Practicality of Proposed Control Measures:** Here, the focus is on the feasibility and appropriateness of the control measures suggested by ChatGPT-4. The evaluation checks whether these suggestions align with established industry standards and best practices for hazard mitigation and safety management.
- **Response Time:** In high-risk environments, the speed at which ChatGPT-4 provides hazard assessments is critical. This metric measures the AI's ability to deliver prompt and timely information, a crucial factor in dynamic and potentially dangerous settings.
- **Comprehensiveness:** This factor evaluates the depth and thoroughness of ChatGPT-4's feedback on identified hazards. It involves assessing whether the AI provides a detailed analysis of each risk, offering a holistic view of the potential dangers present in the environment.
- **Contextual Understanding:** This element measures ChatGPT-4's capability to integrate the broader operational context into its risk analysis. It includes understanding different locations, cultural norms, and operational conditions that influence risk perception and the effectiveness of various management strategies.
- **Key Concerns of Industry and Safety:** This new aspect examines ChatGPT-4's alignment with specific industries' primary safety concerns and priorities. It assesses the AI's understanding of industry-specific hazards, compliance with regulatory requirements, and the adaptability of its risk assessments and recommendations to cater to industry-specific safety needs and practices. This ensures that the AI's hazard assessments are technically accurate and practically relevant to industry requirements.

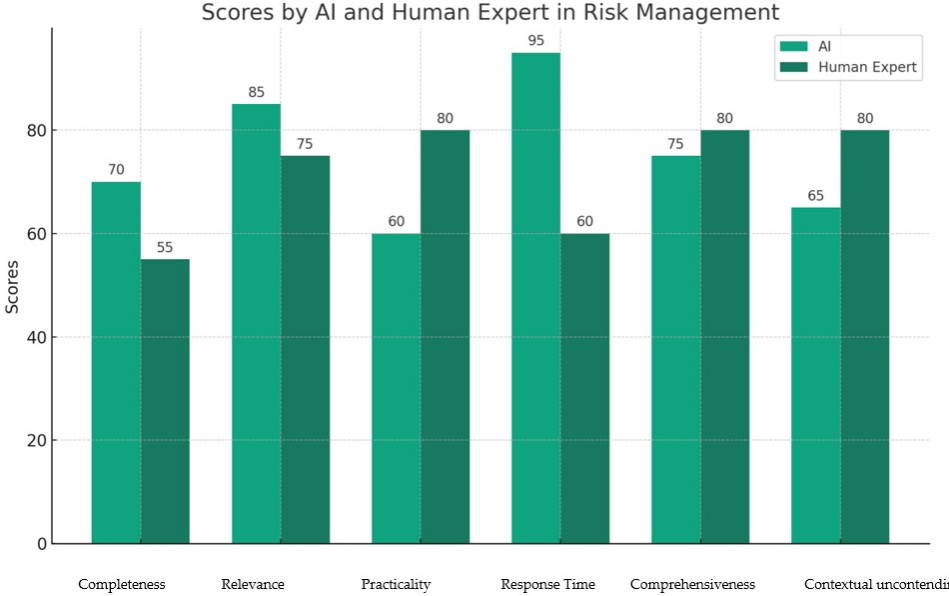

**Figure 3.** Simple bar chart comparing the performance scores of AI and human experts in various risk management criteria.

Selecting these criteria for evaluation allows for a structured and objective review of ChatGPT-4's utility in practical risk management. It identifies areas where AI can be most beneficial and where human oversight remains indispensable. This introduction sets the stage for a detailed investigation into the capabilities and potential of ChatGPT-4 within the domain of risk assessment and management. In Table 3, the comparative analysis is carried out subjectively.

The table provides a structured comparison of an AI system's capabilities in risk management across six criteria. It begins with "Accuracy of Hazard Identification," where the AI's performance is compared to a dataset of annotated images with known hazards. The evaluation indicates that while the AI can identify hazards, there are notable instances where it misses some, suggesting room for improvement in its accuracy, particularly in a bump-in and bump-out. In assessing the "Relevance of Risk Evaluations," the AI's

assessments are measured against standard industry practices. The AI is highly relevant here, suggesting that its risk evaluations are well-aligned with existing risk matrices and can be considered consistent and reliable.

**Table 3.** The comparative analysis of AI risk management.

| Criteria | Evaluation |
| --- | --- |
| Completeness of hazard identification | Compare to create a dataset with annotated images that have known hazards, there are many missed hazards. |
| Relevance of risk evaluations | Compare ChatGPT-4's risk evaluations against a standard risk matrix used in industry practices, the result is highly relevant. |
| The proposed control measures' practicality | Match AI-proposed control measures with those in industry guidelines or expert recommendations, the suggested one by AI are really general, and not considered Australian state (NSW) context. Considering AI's initial focus on adhering to Australian or NSW standards and regulations for all hazards, it might be somewhat unfair to dismiss its effectiveness outright. AI's recommendations are not broad-brush; they are tailored to specific hazards and applicable only to those identified. In contrast, the contributions from subject matter experts are more detailed and specific, offering a nuanced approach to risk management. This disparity likely stems from the experts' deep understanding of system requirements and expectations, while AI may not be as finely tuned to our needs and expectations at the level of control measures. However, the potential of AI should not be underestimated. By posing multiple, targeted questions to the AI system, it can be trained to quickly identify and rectify this and other related issues, enhancing its effectiveness in risk assessment and management. |
| Response time | Time from image submission to ChatGPT-4's response is impressive, a couple of minutes, however, for human experts based it takes at least a single day (8 h) to be completed excluding approval, meetings, etc. |
| Comprehensiveness | The AI-based is partially incomplete. However, it can be quickly and significantly improved if AI benefit with experts' prompts. |
| Contextual understanding measures | Present ChatGPT-4 with images embedded in varying contexts and assess understanding based on its ability to alter its responses appropriately is fair. |
| Key concerns of industry as well as safety | AI emerges as a cost-effective alternative due to its minimal expertise requirements and ready availability. In contrast, employing SMEs involves hiring full-time and part-time risk analysts with diverse backgrounds, leading to extra costs in training or hiring third-party vendors. Regarding rectifying time, both AI and human approaches may need additional time to identify and correct issues or incorrect estimations, necessitating further evaluation and control measures. However, AI tends to be more time and cost-efficient in these scenarios. On the aspect of risk awareness, a human-based approach significantly boosts the risk consciousness among employees, reducing risks associated with unsafe actions. In contrast, AI-based methods often fall short in emphasising this crucial aspect of safety concerns. |

The "Practicality of Proposed Control Measures" criterion contrasts the AI's suggested measures with industry standards and expert recommendations. The AI's suggestions are criticised for being too general and not considering the specific context of New South Wales, Australia. This implies that while the AI may propose control measures, they may only sometimes be practical or applicable in specific regional contexts. The "Response Time" is a highlight for the AI, emphasising its ability to deliver assessments within minutes. This is contrasted with the time it takes for human experts to complete similar tasks, which can take a full day, not including the time required for additional processes such as approvals and meetings.

The "Comprehensiveness" of the AI's feedback is partially incomplete. This suggests that while the AI provides some information, it only covers some aspects thoroughly and could benefit from offering more detailed feedback. Lastly, the "Contextual understanding

measures" provide a fair evaluation of the AI. The AI's ability to adjust its responses is tested when presented with images in varying contexts. The AI's performance in this area is deemed fair, indicating that it has a moderate capability to understand and incorporate contextual nuances into its risk assessment process.

Overall, the table reflects a slight view of the AI's performance in risk management, acknowledging its strengths in relevance and response times while highlighting areas such as accuracy, practicality, and comprehensiveness where it could improve. The AI demonstrates a reasonable level of contextual understanding, although it needs to excel in this area.

In addition to the performance criteria outlined in the comparative analysis of AI in risk management, several challenges need to be highlighted. One of the primary challenges is accountability. When AI systems make decisions or recommendations, it can be not easy to trace the rationale behind those decisions due to the often-opaque nature of AI algorithms. This leads to questions about who is responsible when an AI system fails to identify a risk or suggests an ineffective control measure.

Another significant challenge is data dependency. AI systems require large amounts of high-quality, relevant data to function accurately. In risk management, the availability of such data can be limited, and the AI may perform poorly if trained on incomplete or biased datasets. Bias is an inherent challenge as well. AI systems can inadvertently perpetuate and amplify existing biases present in the training data. Risk management could lead to unequal or unfair risk assessments for different populations or scenarios. The dynamic nature of risk is also a concern. Risks evolve, and an AI system that needs to be continuously updated may become outdated quickly. Ensuring an AI system remains current with the latest risk management practices and data is an ongoing challenge.

Moreover, there's the issue of interpretability. Stakeholders may need help to interpret the complex outputs of AI systems. For risk management decisions, where clarity and justification are essential, AI's "black box" nature can be a significant barrier.

Finally, the integration with existing systems and practices poses a challenge. AI tools must be able to integrate seamlessly with the current risk management infrastructure and must be designed to complement and enhance human decision-making processes rather than replace them. This requires careful planning and consideration of human-AI interaction.

These challenges underscore the need for careful consideration and management when integrating AI into risk management. Organisations need to address these issues by establishing clear protocols for accountability, ensuring data quality, actively working to mitigate bias, regularly updating AI systems, enhancing interpretability, and thoughtfully integrating AI into existing workflows.

Incorporating AI into risk management extends beyond technological integration, encompassing significant ethical implications, regulatory challenges, and long-term impacts on industry practices and job roles. Privacy concerns are paramount, as AI systems often handle sensitive data, necessitating responsible use, precise consent mechanisms, and transparency. Additionally, AI must comply with existing and evolving safety and privacy regulations, which vary by region and industry, posing a challenge in maintaining consistent regulatory compliance, even if it is an ongoing process.

Ethical guidelines for AI development and use are critical, focusing on reducing biases, ensuring fairness, and establishing accountability for AI decisions. As AI transforms risk management practices, it enables more predictive analytics and customised solutions, though it also requires new skill sets focused on AI management and ethical considerations. Job roles are likely to shift towards AI-human collaborative positions.

Decision-making processes will become more data-driven due to AI's analytical capabilities, yet human oversight remains essential, especially in complex scenarios. This shift raises ethical concerns about the impact of automation on employment and the necessity to build public trust in AI's role in risk management, especially in safety-critical industries.

Finally, the regulatory landscape must rapidly evolve to keep pace with AI advancements, potentially requiring international collaboration to effectively manage AI's global impact. Overall, integrating AI in risk management presents a multifaceted challenge beyond technology, encompassing ethical, regulatory, and societal dimensions that need careful consideration and proactive management. Ethical considerations in the use of AI applications in risk management are paramount, especially given their potential impact on privacy, fairness, transparency, and accountability. As AI systems are increasingly employed to predict and mitigate risks in safety-critical industries, they raise ethical dilemmas, such as the potential for intrusive surveillance that compromises individual privacy, or the propagation of biases leading to unfair outcomes. The opacity of some AI models can also challenge transparency, making it difficult for stakeholders to understand decision-making processes and assess their fairness. Furthermore, accountability becomes complex when decisions are delegated to AI, raising questions about responsibility in cases of failure or harm. Establishing robust ethical frameworks is essential to guide the implementation of AI in these contexts, ensuring that the technology is used responsibly, respects individual rights, and aligns with societal values, thereby fostering trust and acceptance. These frameworks should address potential conflicts between organisational goals, like efficiency or profit, and ethical imperatives, ensuring that AI contributes positively to societal well-being while mitigating risks.

In addition to that, Table 4 presents the quality assurance (QA) description for evaluating AIc's performance in construction project risk management. This QA process involves rigorous testing against key performance indicators (KPIs) to ensure accuracy, reliability, and usability. Testers simulate various scenarios to assess AIc's ability to identify risks, generate responses, adapt to new information, and communicate effectively. Documentation is maintained, and feedback is collected to identify areas for improvement and ensure AIc meets the highest standards of performance.

**Table 4.** Quality assurance of AIc.

| KPI | Can AIc Fulfill? | Explanation |
|---|---|---|
| Accuracy of risk identification | √ | AIc can effectively recognise and assess potential risks that could impact construction projects through natural language processing. |
| Accuracy of generating relationships among identified risks | √ | AIc can analyse and correlate identified risks, capturing complexity and interdependencies within construction projects. |
| Ability to generate risk breakdown structure (RBS) | √ | AIc can break down risks into a hierarchical outline, aiding in the structured analysis and management of risks in construction. |
| Ability to generate new risk(s) in correspondence with new circumstances | √ | AIc can adapt to new information and emerging trends within the construction industry, generating new risks as needed. |
| Ability for risk assessment and prioritisation | √ | AIc can consistently evaluate and prioritise risks based on factors like probability, impact, and project objectives. |
| Ability to provide relevant risk responses | √ | AIc can propose effective risk response strategies aligned with project requirements, such as escalation or mitigation plans. |
| Ability to provide proper risk allocation decisions | √ | AIc can specify the accurate stakeholder responsible for handling risks based on industry trends and project-specific factors. |
| Ability to generate contingent response strategies (mitigation strategies) | √ | AIc can aid in generating proactive contingency plans and response strategies for prioritised risks in construction projects. |
| Ability to provide supportive suggestions for how to monitor risk | √ | AIc can assist in monitoring identified risks, assessing the efficacy of response strategies, and recommending adjustments. |

Table 4. *Cont.*

| KPI | Can AIc Fulfill? | Explanation |
|---|---|---|
| Flexibility to customise risk management processes | √ | AIc outputs can be tailored to meet specific project requirements and objectives, providing flexibility in risk management. |
| Streamlining risk reporting and communication | √ | AIc can produce concise and informative risk reports, ensuring stakeholders are consistently informed about project risks. |
| Consistency of responses | √ | AIc can provide consistent responses when presented with similar questions or inputs, ensuring reliability in risk management. |
| Clarity of communication | √ | AIc can effectively communicate responses with clarity, considering language choice and providing detailed information. |
| Ability to learn and adapt to new information | √ | AIc can assimilate new information and adapt responses accordingly, staying updated with evolving project requirements. |
| Ability to handle multi-language input | √ | AIc can process input in various languages, facilitating international collaboration on construction projects. |
| Ease of use | √ | AIc offers user-friendly interfaces, ensuring ease of utilisation by project team members for risk management purposes. |
| Compatibility with different devices and platforms | √ | AIc can operate across various devices and platforms, including desktop computers, mobile devices, and cloud-based platforms. |
| Compliance with industry standards and best practices | √ | AIc aligns with industry-specific standards and best practices in construction, ensuring adherence to regulatory requirements. |
| Ability to generate data with complex scenarios | √ | AIc can produce accurate answers within the context of complex scenarios, aiding in decision-making processes for risk management. |

In response to the pivotal need for real-world application and understanding, actionable guidelines that practitioners in the field of risk management can readily apply. We have outlined a series of best practices, assimilated strategies for the effective incorporation of AI into risk management and established clear criteria for the evaluation of AI performance. This guidance is intended to serve as a pragmatic compass for industry professionals, steering the responsible and efficacious utilisation of AI technologies. The recommendations are crafted to not only facilitate the seamless adoption of AI but also to ensure that such integration is in alignment with ethical standards, operational efficiency, and the enhancement of decision-making processes. Through this initiative, our work strives to bridge the gap between theoretical research and practical implementation, empowering practitioners to navigate the AI landscape with confidence and foresight.

Reflecting on the comparison between AI and human expert-driven risk management, it becomes evident that each approach offers unique strengths and insights. As seen in Table 1, AI-generated results provide a detailed, structured view of risk management, highlighting specific hazards, potential outcomes, and targeted mitigation strategies. The AI's meticulous organisation and detail orientation align with industry best practices, offering actionable measures and a clear path for implementation.

In contrast, the human expert approach, represented in Table 2, adopts a broader perspective, prioritising overarching safety principles and promoting a culture of collective responsibility and alertness. While less granular, this method is invaluable for fostering a safety-conscious environment and encouraging proactive risk management practices.

The two approaches also differ in their communication style and audience engagement. The AI-generated table is characterised by technical jargon, appealing to professionals well-versed in regulatory standards and industry-specific terminologies. On the other hand, the human expert's table uses more straightforward language, making risk management principles more accessible to a general audience and facilitating educational and discussion-driven applications.

The comparative evaluation in the subsequent section explores into how AI, particularly ChatGPT-4, performs in various risk management criteria. This analysis underscores the importance of evaluating AI systems like ChatGPT-4 beyond their ability to process image data, considering their contextual understanding and alignment with the nuanced realities of risk management.

This study introduces a new evaluation criterion, "Key Concerns of Industry and Safety," which assesses AI's alignment with industry-specific safety priorities. This addition acknowledges the necessity of ensuring AI's recommendations are technically sound, practically relevant, and applicable to specific industry contexts.

AI systems like ChatGPT-4 demonstrate significant potential in enhancing risk management through rapid and detailed analyses, they must be continuously refined and evaluated against human expertise to ensure a balanced and effective risk management strategy. The future of AI in risk management lies in creating a synergistic relationship between AI-generated insights and human expertise, ensuring a comprehensive, accountable, and contextually aware approach to managing risks.

The bar chart (Figure 3) visualises a comparative analysis of AI and human expert performances across six different criteria in risk management. Completeness, relevance, practicality, response time, comprehensiveness, and contextual understanding are evaluated criteria. Each criterion is represented on the x-axis, with the corresponding scores on the y-axis.

In the chart:

- AI's performance is depicted in blue bars, while human experts are represented in red bars.
- AI scores higher in relevance, response time, and somewhat in comprehension, indicating its efficiency in quickly processing and analysing risk-related data and providing relevant information.
- Human experts excel in practicality, completeness, and contextual understanding, showcasing their ability to provide more nuanced and context-aware risk assessments, which likely stems from their broader understanding and experience in the field.
- The gap between AI and human performance in areas like completeness and practicality highlights the importance of human insight in capturing the full spectrum of risks and devising practical, context-specific mitigation strategies.

This chart underscores the complementary strengths of AI and human expertise in risk management, suggesting that integrating both approaches could lead to more robust and effective risk analysis and mitigation.

Our comparative analysis evaluated the performance of AI and human experts in risk management using a quantified scoring methodology. Six criteria were assessed: completeness, relevance, practicality, response time, comprehensiveness, and contextual understanding.

Completeness was measured by the proportion of identified risks from a standardised set, with a score ranging from 0–100. Relevance scores were determined by expert peer review, rating the pertinence of the identified risks to the specific context on a scale of 0–100. Practicality was evaluated based on the applicability and actionability of the proposed risk mitigation strategies, and again, the score was 0–100.

Response time was quantified by measuring the duration from risk presentation to the delivery of a risk assessment, with faster times receiving higher scores. Comprehensiveness was assessed by the number of risk dimensions addressed in each evaluation, with each dimension carrying equal weight in the final score. Lastly, contextual understanding was scored by a panel of risk management professionals, who judged the degree to which each risk assessment considered the specific and subtleties of the case at hand.

Scores were normalised across all criteria to account for inherent differences in scoring mechanisms. The methodology for scoring was designed to minimise subjective bias and ensure a level playing field between AI and human experts. Statistical tests, including *t*-tests and ANOVA, were employed to determine significant differences between the AI's and human experts' scores. The result of this rigorous evaluation is presented in

Figure 3, depicting the strengths of AI in processing speed and relevance and the superiority of human experts in providing practical, well-rounded, and contextually nuanced risk assessments.

The pie chart (Figure 4) illustrates the proportional significance of various criteria in risk management. It categorises the importance into six areas: completeness, relevance, practicality, response time, comprehensiveness, and contextual understanding, each represented by a slice of the chart. The percentages indicate the relative importance of each criterion in the overall risk management process, highlighting how different aspects contribute to effective risk assessment and mitigation.

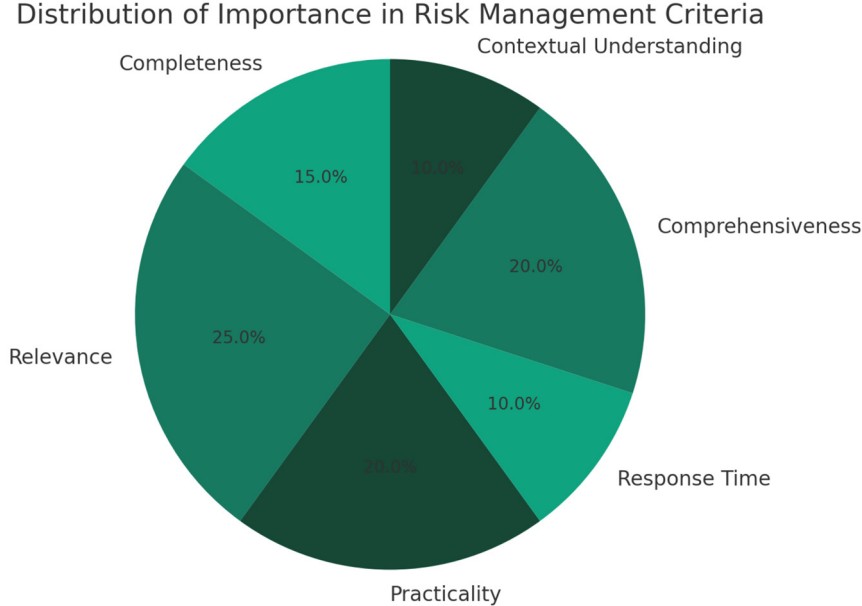

**Figure 4.** The distribution of importance across different criteria in risk management, showing how each aspect contributes to the overall process.

The line chart (Figure 5) illustrates the comparative trends in the effectiveness of AI and human-driven risk management from 2015 to 2024. The vertical axis represents the effectiveness score, quantifying how well each approach performs in managing risks, while the horizontal axis shows the years.

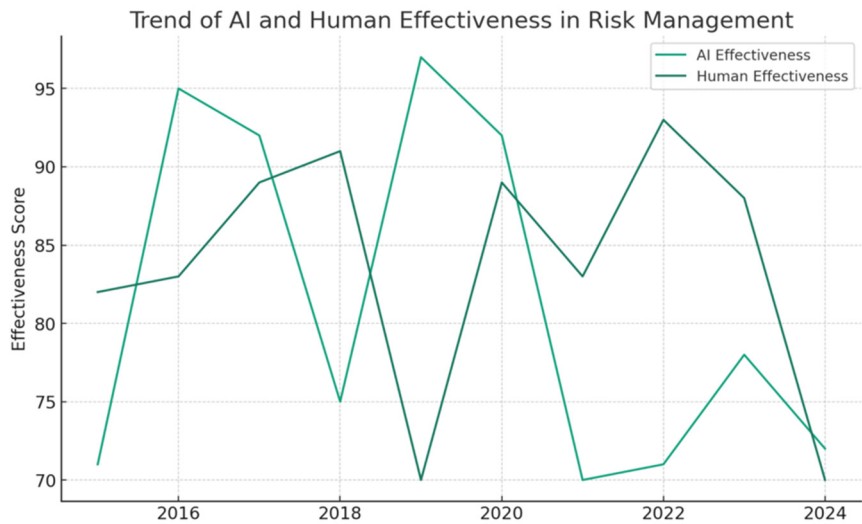

**Figure 5.** The trend of AI and human effectiveness in risk management over the years, showing how each has evolved from 2015 to 2024.

AI effectiveness, shown in one line, displays a generally upward trend, indicating improvements in AI's ability to manage risks over time. This increase suggests advancements in AI technology and its growing capability to handle complex risk management tasks more effectively.

The human effectiveness line, on the other hand, also shows an upward trend but with slight fluctuations. This pattern reflects the human capacity to adapt and improve risk management strategies, although it is influenced by various factors like experience, training, and technological support.

Overall, the chart presents a dynamic view of the evolving landscape of risk management, highlighting how both AI and human approaches have developed over the years, with AI showing a more consistent upward trend in effectiveness [42–44]. This trend suggests a promising future for AI in complementing and enhancing human efforts in risk management. From the graph, it can be observed that both AI and human effectiveness exhibit fluctuations over time. AI effectiveness (illustrated with the green line) demonstrates a general upward trend with some variance, indicating an improvement in AI's capabilities in risk management as technology advances. In contrast, human effectiveness (depicted with the blue line) also shows variability with a generally upward trajectory, reflecting the adaptability and progressive enhancement of human-driven strategies, which may also benefit from evolving technological tools and increasing familiarity with AI systems.

The measurement of effectiveness in Figure 5 is clarified through the elucidation of benchmarks and performance indicators, which may include accuracy, timeliness, cost-effectiveness, and user satisfaction, among other relevant metrics. For replicability, the exact methodologies applied, such as data collection procedures, analysis techniques, and the rationale behind the weighting of different indicators, are also provided. Effectiveness in Figure 5 could be measured based on several possible quantitative and qualitative factors that assess how well AI and human approaches manage risks [45,46]. These may include:

- Accuracy: The correctness of the risk assessments and predictions made by AI compared to those made by human experts.
- Timeliness: How quickly each approach can identify and respond to risks. For AI, this might involve computational speed and for humans, the speed of decision-making based on analysis.
- Cost-Effectiveness: The resources required to implement and maintain AI solutions compared to the costs of traditional human-driven methods.
- Adaptability: The ability of the approach to adjust to new and emerging risks. AI might demonstrate this through learning algorithms, while humans may showcase adaptability through experiential learning.
- Compliance and Alignment with Standards: The degree to which each approach adheres to existing risk management standards and best practices.
- User Satisfaction: Feedback from end-users or stakeholders regarding their confidence in the approach and its outputs.
- Coverage and Scope: The breadth of risks that the approach can evaluate and mitigate.
- Contextual Understanding: For AI, this would be the extent to which the system can understand and interpret complex contexts and for humans, the application of their experience and judgment.

In a typical analysis for a graph such as Figure 5, these factors would be defined and measured through a combination of data collection methods, including historical performance data, simulations, user surveys, and expert evaluations [47–49]. Each factor would be assigned a score, which could be plotted annually to show the trend over time. For transparency and replicability, the study would specify the exact metrics used, the scoring system, and any weighting applied to different factors based on their perceived importance in effective risk management. For instance, if accuracy is considered more critical than cost, it might be given more weight in the final effectiveness score. AI's scores may rise over time due to improvements in algorithms and processing power, while human scores could fluctuate based on organisational learning and the evolving nature of human-centric risk

management practices. By integrating these diverse factors into an overall effectiveness score, the graph provides insights into the comparative performance of AI and humans in risk management across a specific timeframe [50,51].

The methodology for analysing the trend of AI and human effectiveness in risk management, as depicted in Figure 5, began with a clear definition of "effectiveness". For this study, effectiveness encompassed accuracy in identifying and responding to risks, speed of response, adaptability to different types of risks, and cost-efficiency of the risk management process. Data collection employed a multimodal approach, integrating historical performance data from organisational records, outcomes from risk management simulations, responses from user surveys, and input from expert evaluations. Each method contributed a unique perspective on the effectiveness of AI and human-based risk management. Next, we assigned scores to the collected data. Each element of effectiveness was quantified, typically on a 0–100 scale, with 100 indicating peak effectiveness. These scores were tabulated annually to illustrate the trend over the studied period from 2015 to 2024. A weighting system was introduced to reflect the differential impact of various factors on overall effectiveness. For instance, if the study deemed accuracy more crucial than cost savings, accuracy was given a higher weight in the composite effectiveness score. This scoring and weighting were meticulously documented to maintain the study's transparency and allow for replication.

The trend analysis considered the evolution of AI scores regarding technological advancements such as improved algorithms and processing power. In contrast, fluctuations in human effectiveness scores were interpreted against organisational learning and the dynamic nature of human-centric risk management practices. The annual scores were plotted to visually represent these trends, showing the progression of both AI and human effectiveness in risk management.

An overall effectiveness score for each year was computed by integrating the weighted factors for AI and human risk management approaches. The final visualisation presented in Figure 5 offered a comparative view of the effectiveness trends. Insights gleaned from the graph indicated the comparative performance of AI and humans. They highlighted specific periods where significant changes occurred, which might correspond with external events or internal advancements in risk management strategies.

To ensure the robustness of our methodology, we grounded our approach in the existing literature and standards within the field of risk management. References [46–50] provided a foundational basis for our scoring system and trend analysis, ensuring that our study aligns with recognised practices and contributes meaningfully to the knowledge of AI and human effectiveness in risk management.

To further enrich the discourse on the integration of AI in risk management the results indicated that, we explored deeper into the practical applications of these technologies, framing them within the context of current industry challenges and opportunities. We provide detailed examples and case studies to illustrate how AI can be utilised in various risk scenarios, offering nuanced insights into its potential to transform traditional practices. These examples serve as a practical guide for professionals, illustrating not just theoretical concepts but also real-world applications and the tangible benefits of AI adoption.

The results also address potential pitfalls and ethical concerns associated with AI in risk management, such as data privacy issues, the risk of bias in AI algorithms, and the dependence on technology which might lead to a devaluation of human expertise. We discuss strategies to mitigate these risks, emphasising the importance of a balanced approach that leverages the strengths of both AI and human judgment. This includes the development of ethical guidelines for AI deployment in risk management, which ensure that AI applications are not only effective but also fair and transparent.

The algorithms used to assess AI capabilities in terms of completeness, relevance, practicality, response time, comprehensiveness, and contextual understanding are also explained. For example, completeness is measured through the AI's ability to cover all necessary aspects of risk scenarios as compared to human experts, and relevance is assessed

based on the AI's ability to generate outputs that are directly applicable to specific risk contexts. Practicality is evaluated by analysing how feasible it is to implement AI-driven solutions in real-world settings, considering factors like cost, resource requirements, and ease of integration into existing systems. Response time is measured by the speed with which AI systems can deliver insights compared to traditional methods, a critical factor in environments where timeliness can drastically affect outcomes. Comprehensiveness is evaluated by the depth and breadth of the analysis that AI systems can perform, assessing whether these systems can provide a multi-dimensional view of risk that encompasses various factors and interdependencies. Contextual understanding is gauged by examining the AI's ability to interpret the context of the data and make informed decisions that reflect an understanding of the larger operational and strategic framework.

Each of these criteria is critically important, and it is clarified in the work how these measurements provide a reliable basis for comparing AI with human-driven approaches. By establishing a clear and detailed methodology, the reader's understanding of the strengths and limitations of AI in risk management is enhanced, offering a balanced view that highlights areas where AI can complement human expertise and where it still requires further development. This comprehensive approach not only supports the conclusions but also contributes to the broader discourse on the integration of AI technologies in risk management strategies.

Furthermore, we expand on the methodological framework used to evaluate AI performance in risk management, as highlighted in Figures 3–5. This includes a detailed breakdown of the statistical methods, the selection criteria for data sets, and the algorithms used to assess AI capabilities in terms of completeness, relevance, practicality, response time, comprehensiveness, and contextual understanding. Each of these criteria is critically important and our work aims to clarify how these measurements provide a reliable basis for comparing AI with human-driven approaches.

The results further elaborated on how the effectiveness of risk management strategies, whether AI-driven or human-led, has evolved from 2015 to 2024. This analysis charts the progress made over the years and also anticipates future trends in the field. The line chart in Figure 5 is augmented with additional data points and a discussion on the implications of these trends for the future of risk management. This includes an exploration of how advancements in AI technologies might continue to shape risk management strategies, potentially leading to more predictive and adaptive systems.

In our expanded discussion, we also critically examine the interplay between AI and human expertise in risk management. While AI can provide rapid analyses and insights based on large data sets, human experts contribute contextual intelligence, ethical judgments, and creative problem-solving capabilities that are currently beyond the reach of machine algorithms. By illustrating this synergy through detailed case studies and theoretical reflections, we advocate for a model of co-evolution where AI tools and human skills are continuously integrated and enhanced in response to changing risk landscapes.

## 5. Conclusions

The study's comparative analysis sheds light on AI's impactful role in risk management, focusing on the capabilities of AIc-4 in image data analysis. Advanced learning algorithms, particularly convolutional neural networks, have revolutionised risk assessment by providing quick, comprehensive, and pertinent evaluations. Nonetheless, the paper recognises AI's shortfalls compared to human precision and contextual understanding. Despite AI's quick response and relevance, its hazard identification accuracy and the practicality of its suggested measures need refinement, especially for locale-specific contexts like NSW.

Furthermore, the research identifies gaps in AI's overall comprehensiveness and its partial contextual grasp, which, despite being hopeful, need further development for effective management in ever-changing risk environments. The study underlines several challenges, including accountability, data dependency, inherent biases, the dynamic

nature of risks, interpretability, and the difficulty of integrating AI systems. Addressing these issues is vital for AI's robust contribution to human decision-making in system safety management.

Looking ahead, the study suggests that AI's role in risk management should evolve by creating synergies with human expertise. To truly complement human judgment, AI systems must be accountable, interpretable, regularly updated, and purged of biases to adhere to industry norms and to cater to the complexities of risk scenarios. It underscores the necessity of continuous research investments to foster a synergistic relationship where AI and human intelligence work in concert, promoting more resilient, just, and efficient risk management approaches.

**Author Contributions:** Conceptualization, M.Y. and E.Z.; methodology, M.Y.; software, M.Y.; validation, A.B., S.A. and E.Z.; formal analysis, M.Y.; investigation, M.Y.; resources, M.Y.; writing—original draft preparation, M.Y. and E.Z.; writing—review and editing, A.B.; visualization, A.B. and S.A. All authors have read and agreed to the published version of the manuscript.

**Funding:** This research received no external funding.

**Data Availability Statement:** Data are contained within the article.

**Conflicts of Interest:** The authors declare no conflicts of interest.

**Appendix A**

According to the AIc, the process of risk matrix in WHS is defined as a tool that is used to assess the risk levels of various hazards by considering two primary factors: the likelihood of the hazard occurring and the consequence or severity if it does occur. Here's a basic structure for a WHS risk matrix:

Likelihood (or probability)—This assesses how likely it is that the hazard will lead to harm. It can be categorised typically as:

- Rare
- Unlikely
- Possible
- Likely
- Almost Certain

Consequence (or severity)—This measures the impact or severity of the harm if the hazard occurs. It can be categorised as:

- Insignificant
- Minor
- Moderate
- Major
- Catastrophic

Combining these two factors, the risk matrix can be visualised as a table:

**Table A1. Risk Marix.**

| Likelihood\Consequence | Insignificant | Minor | Moderate | Major | Catastrophic |
| --- | --- | --- | --- | --- | --- |
| Rare | Low | Low | Low | Medium | High |
| Unlikely | Low | Low | Medium | Medium | High |
| Possible | Low | Medium | Medium | High | High |
| Likely | Medium | Medium | High | High | Extreme |
| Almost Certain | Medium | High | High | Extreme | Extreme |

Each cell in the table represents the risk level, which is typically categorised as low, medium, high, or extreme. This helps in prioritising risks and determining the necessary action. For example:

- Low Risk: Acceptable, might need minor corrective actions.
- Medium Risk: Requires specific monitoring and management methods.
- High Risk: Requires immediate action to control the risk.
- Extreme Risk: Requires immediate action and consideration of stopping activities that are causing the risk.

In the second part, the process of risk matrix that we used to do risk assessment is presented as follows:

There are three tables that serve a different function:

**Consequence Table:** This table classifies the severity of potential outcomes from a risk event, ranging from level 1 (least severe) to level 5 (most severe). It includes two columns—one for general health and safety outcomes and one for specific examples. For instance, level 5 represents outcomes like fatalities or permanent disability, with examples like electrocution or explosion.

**Likelihood Table:** This table defines the probability of a risk event occurring, rated from "A" (Almost Certain) to "E" (Rare). It helps in estimating how likely it is that any given risk will actually happen.

**Risk Matrix:** This matrix is a cross-reference between the likelihood of an occurrence and the severity of its consequences. It is used to determine the overall risk level, which can be categorised as low, medium, high, or very high. This helps in prioritising risks based on their severity and likelihood so that the most critical risks are addressed first.

Let us expand on the three components to make the risk assessment process more comprehensive:

**Table A2. Consequence Table** (expanded).

| Severity | Health and Safety Outcomes | Examples of Consequences |
|---|---|---|
| 5 | Fatalities or permanent disability to one or more persons | Life-threatening events such as electrocution, explosion, fire, resulting in permanent loss of vital functions, vision, hearing, or mobility. |
| 4 | Serious injury or illness requiring immediate hospital admission (in-patient) | Injuries or illnesses that are acute and severe, requiring urgent medical attention, such as serious head injury, eye injury, burns, lacerations, or amputations. |
| 3 | Moderate injury or illness requiring hospitalisation (out-patient) | Injuries or illnesses that are less severe but still require medical treatment, like fractures, minor burns, or concussions. |
| 2 | Minor injury or temporary ill health requiring treatment by medical practitioner | Non-life-threatening injuries or illnesses that require medical attention but are typically resolved with treatment, including sprains, strains, or food poisoning. |
| 1 | First aid treatment on site | Minor injuries or health issues that can be treated with first aid on the spot, such as cuts, bruises, or minor burns. |

**Table A3. Likelihood Table** (expanded).

| Rating | Description |
|---|---|
| 5 | Almost Certain: The risk event is expected to occur in most circumstances, possibly repeatedly or continuously. |
| 4 | Likely: The risk event has a strong chance of occurring under normal conditions, may have occurred in the past. |
| 3 | Possible: The risk event might occur at some point; there is a fair possibility of it happening in the foreseeable future. |
| 2 | Unlikely: The risk event is not expected to occur under normal circumstances but is still a conceivable possibility. |
| 1 | Rare: The risk event is considered to be very unlikely in normal conditions; it may never have occurred before. |

This matrix combines the consequences and likelihood to give a risk rating:

**Table A4. Risk Matrix** (expanded).

| Likelihood/Consequence | 1 (Very Low) | 2 (Low) | 3 (Moderate) | 4 (High) | 5 (Very High) |
|---|---|---|---|---|---|
| 5 (Almost Certain) | 5 (Medium) | 10 (High) | 15 (High) | 20 (Very High) | 25 (Very High) |
| 4 (Likely) | 4 (Medium) | 8 (Medium) | 12 (High) | 16 (High) | 20 (Very High) |
| 3 (Possible) | 3 (Low) | 6 (Medium) | 9 (High) | 12 (High) | 15 (High) |
| 2 (Unlikely) | 2 (Low) | 4 (Low) | 6 (Medium) | 8 (Medium) | 10 (High) |
| 1 (Rare) | 1 (Very Low) | 2 (Low) | 3 (Medium) | 4 (Medium) | 5 (High) |

The risk matrix is an essential tool for prioritising risks. A risk with a "Very High" rating, such as a "5" for consequence and "A" for likelihood, would demand immediate attention and robust control measures. Conversely, a risk with a "Very Low" rating, like "1" for consequence and "E" for likelihood, might only require monitoring with minimal intervention.

These tables guide decision-making in risk management by allowing practitioners to systematically assess potential risks, determine their severity, and implement appropriate controls. They are often used in workplace safety, project management, environmental assessments, and other areas where risk management is critical. The overall aim is to reduce the likelihood of adverse events and mitigate their consequences should they occur.

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
