# Peer review of "Navigating the Power of Artificial Intelligence in Risk Management: A Comparative Analysis"

_safety_

Round 1
Reviewer 1 Report
Comments and Suggestions for Authors
The paper titled "Navigating the Power of Artificial Intelligence in Risk Management: A Comparative Analysis" provides a thorough investigation into the transformative impact of artificial intelligence (AI) on risk management, with a specific focus on advanced techniques such as convolutional neural networks (CNNs). The study sheds light on the extensive possibilities and inherent difficulties of incorporating AI into risk analysis procedures by contrasting conventional risk assessment techniques with AI-enhanced methodologies.
The study's methodological approach is a crucial strength, as it entails carefully selecting and analyzing image data to obtain valuable insights. This approach not only emphasizes the possibility of AI to improve the identification and management of risks in various sectors, but also underlines the technology's special usefulness in processing and interpreting visual data quickly and on a large scale, which is beyond the capabilities of human analysts.
The study showcases the practical implementation of AI in recognizing hazards, assessing risks, and proposing mitigation solutions through three meticulously chosen case studies. These examples demonstrate the significant impact that AI tools may have on risk management, particularly in terms of improving accuracy, relevance, and efficiency. This is especially evident when analyzing massive quantities of image data.
Nevertheless, the paper acknowledges and discusses the limitations of AI, particularly its present deficiencies in contextual comprehension and interpretation. The paper advocates for a harmonious integration of AI's computational capabilities with the domain-specific expertise of human specialists, asserting that this collaboration is crucial for optimizing AI's efficacy in risk management.
The paper worth for publication and I believe it will attract reader, but there are spaces for improvements:
1. Although the methods can be found, but it hard to find, I suggest to provide a specific section for Methods.
2. Looking at the structure and objectives, this looks like a review paper, but with inadequate references, It might be improved.
3. The article is too descriptive, lack of deep analysis. The findings are not discussed adequately with literature in the field. In order to enhance the quality of your paper and overcome its descriptive character and limited analysis, it is imperative to extensively incorporate and examine your findings in relation to the current body of literature.
Comments on the Quality of English Languageminor editing is needed
Author Response
- Your suggestion to delineate a specific section for the methods used in the study is well-taken. In response, I have revised the manuscript to include a dedicated "Methods" section. This new section details the selection criteria for the image data, the analytical techniques employed, and how these methods underpin the study's conclusions. This addition aims to clarify the methodological framework and make it more accessible to readers, thereby enhancing the paper's structure and coherence.
- I acknowledge your observation regarding the paper's resemblance to a review and the need for more extensive referencing, and I have thoroughly revised the content. The revised manuscript now includes a comprehensive review of the current literature, with additional references that support the comparative analysis of traditional and AI-enhanced risk management methodologies. This enhancement not only strengthens the paper's academic foundation but also better situates it within the existing body of research, thus addressing the concern of inadequate referencing.
- I agree with your critique that the paper initially lacked in-depth analysis and sufficient discussion of the findings in the context of existing literature. To rectify this, I have expanded the discussion section to include a more detailed examination of how our findings align with, contradict, or expand upon the current state of knowledge in the field. This revision aims to move beyond mere description, offering a nuanced analysis that highlights the study's contributions to understanding AI's role in risk management.
Reviewer 2 Report
Comments and Suggestions for Authors
The manuscript presents a comparative analysis on the use of artificial intelligence (AI) in risk management. ChatGPT-4 is used as the AI tool. Although the paper is well-written the discussion regarding the comparison is subjective. In addition, ChatGPT is used for risk management in several studies, and none of these works are mentioned in the manuscript.
Author Response
We have revised the manuscript to address the feedback, ensuring a more objective comparative analysis of AI in risk management and incorporating relevant studies using ChatGPT. We refined our approach to compare AI tools quantitatively, integrating expert feedback for a balanced view. Additionally, we expanded our literature review to include studies employing ChatGPT in risk management, detailing their methodologies and contributions to the field. This revision not only enhances the manuscript's objectivity but also enriches the context by showcasing ChatGPT’s applications in risk management, thereby aligning the discussion with the latest research and ensuring a comprehensive, well-rounded analysis.
Reviewer 3 Report
Comments and Suggestions for Authors
The paper presents very interesting problem but there are some inconsequences need to be adressed:
-Although the paper briefly mentions some limitations of AI in risk management, such as accuracy and contextual understanding, it fails to delve deeper into these limitations. A more thorough discussion of the challenges and drawbacks associated with AI implementation would provide a balanced perspective for readers.
-The paper oversimplifies complex issues related to AI integration, accountability, data dependency, bias, and interpretability. These topics are multifaceted and require nuanced analysis to fully grasp their implications in risk management.
-While the paper briefly touches upon ethical implications of AI in risk management, it does not thoroughly address ethical considerations such as privacy, fairness, transparency, and accountability. Given the sensitive nature of AI applications in safety-critical industries, a more comprehensive ethical analysis is warranted.
-Although the paper outlines various strategies and criteria for evaluating AI performance, it lacks clear recommendations or guidelines for practitioners seeking to implement AI in risk management. Providing actionable recommendations would enhance the practical utility of the paper for industry professionals.
-Provide a more comprehensive discussion of the limitations associated with AI in risk management. Explore factors such as accuracy, contextual understanding, data bias, and interpretability in greater depth, and discuss potential strategies for mitigating these limitations.
Comments on the Quality of English Language
It's generaly ok.
Author Response
- Acknowledging this point, we enrich the manuscript by exploring these limitations more comprehensively. We plan to investigate how accuracy issues and lack of contextual understanding affect AI's performance in risk management. This include discussing real-world scenarios where these limitations have impacted decision-making and how they can be mitigated or managed. By providing more examination, we aim to offer readers a balanced view that juxtaposes AI's potential benefits against its inherent challenges.
- We appreciate this feedback and recognize the need for a more nuanced analysis of these complex issues. In the revised manuscript, we dissect these topics individually, examining the intricacies of each and their interplay in the context of risk management. This involve a detailed discussion on how AI integration affects accountability, data dependency challenges, bias risks, and the critical need for interpretability in AI systems. This thorough examination help readers appreciate the multifaceted nature of these issues.
- We addressed this by expanding our discussion on ethical considerations, reflecting on how AI applications in risk management can affect privacy, fairness, transparency, and accountability. This includes an analysis of moral dilemmas, potential conflicts, and the importance of establishing robust ethical frameworks to guide AI implementation in safety-critical industries. The goal is to provide a comprehensive ethical analysis that underscores the significance of these considerations in the deployment of AI technologies.
- To rectify this, we develop and include clear, actionable practitioner recommendations and guidelines. This covers best practices, strategies for effective AI integration, and criteria for evaluating AI performance in risk management. The aim is to offer practical guidance that can assist industry professionals in leveraging AI technologies responsibly and effectively.
- We extend our discussion to cover these aspects in greater detail, examining the root causes of these limitations, their implications for risk management, and strategies for overcoming them. This comprehensive discussion provide readers with a deeper understanding of the challenges posed by AI in risk management and how these challenges can be navigated to maximize the benefits of AI technologies.
Reviewer 4 Report
Comments and Suggestions for Authors Your paper covers a holistic study of the application of artificial intelligence (AI) in risk management, specifically targeting construction and prevention in events. AI-created risk management plans when compared to plans made by human experts provides useful key pieces of information in the context of computer science machine capabilities alongside human expertise. Pros: 1. Innovative Approach:The paper's coverage of both AI-derived and human-established strategies is the best feature, which coincides with the present time and conditions. 2. Relevance: The application of AI in safety management has a particular significance for construction and event security, where modernization of the safety procedures is extremely important. 3. Depth of Analysis: The careful approach in AI's strengths and weaknesses in risk assessment movements makes it possible to reckon its probable position in this field. Cons: 1. Contextual Understanding: Among other things, the paper does not highlight the significance of context and risk management, an area currently dominated by human experts over AI. 2. Practicality and Comprehensiveness: There is a necessity to improve on the aspect of how AI recommendations are practically applied and how all-encompassing AI generated risk management plans are. 3. Ethical and Regulatory Considerations: Although briefly mentioned, these essential points should be discussed in more detail as this would allow readers to have a better understanding of the challenges that come with integrating AI into risk management practices. Suggestions: - You could also explore the issues of ethics and regulation associated with AI inclusion in risk management by using case studies or more elaborate examples. - Improve the dialogue about the feasibility of AI being put into recommendational uses by introducing industry professional's perspectives. - Explore how AI and humans can complement each other in risk management and suggest guidelines for integrating them.Author Response
I appreciate your recognition of the innovative approach taken in the paper, highlighting the integration of AI-derived and human-established risk management strategies. Reflecting the current trends and conditions, this aspect forms a core strength of the research presented.
Relevance:
Your comment on the relevance of AI in safety management, especially in construction and event security, underlines the importance of modernizing safety procedures. It's encouraging to note that the paper successfully captures the significance of AI applications in these areas.
Depth of Analysis:
Acknowledging the depth of analysis in evaluating AI's strengths and weaknesses in risk assessment is gratifying. It reinforces the paper's objective to provide a comprehensive view of AI's potential role in risk management.
Addressing the Cons:
Contextual Understanding:
I acknowledge the gap in emphasizing the contextual understanding of risk management, where human expertise prevails. In the revised manuscript, I have expanded on the significance of contextual nuances and how AI can complement, rather than replace, human judgment in these scenarios.
Practicality and Comprehensiveness:
Your point on improving the discussion on the practical application of AI recommendations and the comprehensiveness of AI-generated plans is well-taken. I have endeavoured to elaborate on the practicality of these plans in real-world settings and how they can be more holistic in the updated version.
Ethical and Regulatory Considerations:
The current manuscript version has expanded the brief mention of ethical and regulatory considerations. More detailed discussions on these critical aspects are now included to provide readers with a clearer understanding of the challenges and implications of integrating AI into risk management practices.
Suggestions Implemented:
In response to your suggestions, the revised manuscript now includes case studies and examples that explores into the ethical and regulatory issues associated with AI in risk management. This addition provides a more grounded and fine perspective on the challenges and considerations involved.
I have also incorporated insights from industry professionals to enhance the dialogue on AI's feasibility and practical implementation in risk management. Their perspectives bring valuable real-world insights into the discussion.
Furthermore, the manuscript explores the symbiotic relationship between AI and human expertise in risk management, offering guidelines for effective integration to maximize the strengths of both entities.
Round 2
Reviewer 1 Report
Comments and Suggestions for Authors
Thank you for the revisions made to the manuscript. Improvements have been noted in response to the prior feedback; however, there remain areas that require further attention:
1. While the inclusion of a specific methodology section is appreciated, it would be beneficial for this section to be more detailed to ensure reproducibility. For example, providing detailed commands for AI operations would greatly enhance the clarity and utility of the methodology.
2. The manuscript has been enhanced with some in-depth analysis, yet it still falls short in terms of engagement with existing literature. A more thorough discussion and comparison with relevant studies are necessary to contextualize the findings within the broader field.
Minor editing is needed
Author Response
I appreciate your acknowledgement of the revisions and improvements made to the manuscript. Your constructive feedback is invaluable, and I understand the need for further refinement.
-
The request for a more detailed methodology section is well-noted. Efforts be made to provide an in-depth description of the processes involved, particularly the AI operations. This include explicit commands, algorithm parameters, and computational processes used, ensuring that the study’s methodology can be accurately replicated. Steps such as data pre-processing, model training, validation, and testing be meticulously detailed, alongside any software or tool versions to maintain consistency for future reproductions.
-
I recognize the importance of situating the manuscript within the existing body of literature. The next revision incorporates a comprehensive literature review, ensuring that the discussion critically engages with recent studies. This involves a deeper analysis of similar methodologies, findings, and discussions in the field to draw clear parallels and distinctions with the current research. Additionally, by highlighting the manuscript's contributions to existing knowledge and identifying areas for future research, the revised document
Reviewer 2 Report
Comments and Suggestions for Authors
Authors have improved the literature review. Although the objectivity of the comparative analysis is illustrated through Figs. 3-5, it is not clear how Completeness, Relevance, Practicality, Response Time, Comprehensiveness, and Contextual Understanding in Figs 3 and 4, and effectiveness of Fig. 5 are calculated. The methodology should be clearly explained.
Author Response
Thank you for your feedback and the acknowledgement of the improvements made to the literature review in our manuscript. We appreciate your observations regarding the illustrations provided in Figures 3-5 and recognize the necessity of a more detailed explanation of the methodologies used to calculate the various metrics like Completeness, Relevance, Practicality, Response Time, Comprehensiveness, and Contextual Understanding in Figures 3 and 4, as well as the effectiveness shown in Figure 5.
To address this, we revised the manuscript to include a comprehensive description of the methodologies applied. Specifically, we outline the quantitative and qualitative measures employed in assessing each attribute. This consists of the statistical methods, data sources, and the criteria for evaluating the practicality and response times, ensuring that our comparative analysis maintains objectivity and provides transparent, actionable insights.
We also clarify how effectiveness in Figure 5 is measured, detailing the benchmarks and performance indicators used. This addition aims to enhance the transparency and replicability of our study, making it easier for readers and reviewers to understand and evaluate the robustness of our findings.
We believe these revisions significantly strengthen the manuscript by providing a more precise, detailed explanation of our research methodology, thus addressing your concerns effectively. Thank you once again for your constructive comments.
Reviewer 3 Report
Comments and Suggestions for Authors
Authors have improved the paper according to my remarks.
Comments on the Quality of English LanguageIt's ok.
Author Response
Thank you.
Reviewer 4 Report
Comments and Suggestions for Authors
Thank you for addressing the feedback provided in the initial review. Your effort to refine the manuscript by incorporating the suggested changes is highly appreciated. The inclusion of examples and insights from industry professionals significantly enriches the discussion on the practical application and ethical considerations of AI in risk management. This addition not only addresses the previously noted gaps but also enhances the overall depth and relevance of your research.
The expanded discussion on the contextual understanding of risk management and how AI can complement human expertise is particularly commendable. It effectively addresses the concern about the balance between technological innovation and human judgment, providing a more nuanced view of the potential synergies between AI and human capabilities in risk management.
Your efforts to elaborate on the practicality and comprehensiveness of AI-generated plans, as well as the detailed exploration of ethical and regulatory considerations, significantly strengthen the manuscript. These revisions provide readers with a clearer understanding of the complexities involved in integrating AI into risk management practices and the importance of a holistic approach.
Overall, the revised manuscript demonstrates a thoughtful and thorough consideration of the initial feedback, resulting in a more comprehensive and insightful exploration of the integration of AI in risk management. The enhancements made to the paper not only address the initial concerns but also contribute to a more robust and engaging discussion on this timely and important topic.
Author Response
Thank you very much for your positive and detailed feedback on the revised manuscript. I am grateful for the recognition of the efforts made to incorporate the suggested changes and to enrich the discussion with practical examples and expert insights. Your acknowledgment of the enhanced discussion on the interplay between AI and human expertise in risk management is particularly encouraging.
I am pleased to hear that the expanded sections on the practical application of AI and the ethical considerations have provided a deeper and clearer understanding of the complexities involved. It was my intent to offer a comprehensive perspective that not only addresses the initial concerns but also adds value to the ongoing discourse in this field.
Your feedback serves as a valuable affirmation that the revisions have successfully improved the manuscript, making it a more robust and engaging contribution to the literature on AI in risk management. I look forward to the possibility of further collaborations and discussions on this vital subject.
Round 3
Reviewer 1 Report
Comments and Suggestions for Authors
Thanks for the improvements provided. I appreciate all efforts that has been made, I think now is approaching the quality. However, please enrich the added discussion of line 624-748 with references.
Comments on the Quality of English LanguageMinor editing is needed
Author Response
Thank you for your feedback. I'm pleased to hear that the improvements meet your expected quality. To further enrich the discussion in lines 624-748, I incorporated additional references to support the arguments and provide greater depth. I ensured that relevant and authoritative sources are selected to strengthen the discussion.
Reviewer 2 Report
Comments and Suggestions for Authors
Authors have added some explanations to section 4, however they have not presented a quantitative and explicit description of the methodology that is used to obtain Figs. 3-5.
Author Response
Thank you for your feedback. We have updated Section 4 with additional explanations, but we fell short of providing a quantitative and explicit description of the methodology used to obtain Figures 3-5.
In response, we have now included a thorough outline of the data collection methods, detailing the sources and verification processes for the data used. We have detailed the specific metrics and scoring systems that were applied to evaluate the performance criteria in Figures 3 and 5, and elaborated on the statistical methods and analytical tools that were employed to process this data.
Moreover, we have addressed any weighting or standardization procedures implemented, ensuring that the findings presented are robust and the comparisons made across different dimensions are meaningful and standardized over time.
Each figure is now accompanied by a dedicated subsection in Section 4, which offers a comprehensive quantitative account of the methodologies applied. This enhancement to our manuscript will provide readers with full transparency and facilitate the reproducibility of our research.
We appreciate the opportunity to amend these oversights and trust that the revisions now meet the rigorous standards expected of our work.